# Ultrahigh loading dry-process for solvent-free lithium-ion battery electrode fabrication

Minje Ryu[1], Young-Kuk Hong[1], Sang-Young Lee [1] & Jong Hyeok Park [1] ✉

The current lithium-ion battery (LIB) electrode fabrication process relies heavily on the wet coating process, which uses the environmentally harmful and toxic N-methyl-2-pyrrolidone (NMP) solvent. In addition to being unsustainable, the use of this expensive organic solvent substantially increases the cost of battery production, as it needs to be dried and recycled throughout the manufacturing process. Herein, we report an industrially viable and sustainable dry press-coating process that uses the combination of multiwalled carbon nanotubes (MWNTs) and polyvinylidene fluoride (PVDF) as a dry powder composite and etched Al foil as a current collector. Notably, the mechanical strength and performance of the fabricated $LiNi_{0.7}Co_{0.1}Mn_{0.2}O_2$ (NCM712) dry press-coated electrodes (DPCEs) far exceed those of conventional slurry-coated electrodes (SCEs) and give rise to high loading (100 mg cm$^{-2}$, 17.6 mAh cm$^{-2}$) with impressive specific energy and volumetric energy density of 360 Wh kg$^{-1}$ and 701 Wh L$^{-1}$, respectively.

Rechargeable lithium-ion batteries (LIBs) have become a new energy storage device in various fields owing to the global interest in green technologies and increased awareness of environmental issues[1–5]. Although LIBs are well known as clean energy storage devices, they are yet to become a silver bullet for sustainable development due to the toxic volatile solvent pollution that occurs during the early stage of their electrode fabrication[6–8]. The fabrication of conventional LIB electrodes involves the coating of metallic current collectors with a viscous slurry made by mixing the active material, the conductive agent, and a polymeric binder such as polyvinylidene fluoride (PVDF), in N-methyl-2-pyrrolidone (NMP) solvent[9–11]. However, this expensive organic solvent evaporates very slowly, and its drying and recovery constitute a notable portion (approximately 78%) of the total electrode production cost[12]. Moreover, NMP is not only toxic (known to cause male infertility) but also flammable. Thus, prolonged exposure can be hazardous to the health of the workers as well as can potentially lead to fire hazards[13]. Owing to these alarming issues, many battery researchers and manufacturers are currently working toward eradicating NMP from the electrode fabrication process.

The dry process is considered a new electrode fabrication method for post-LIB electrodes since it offers unparalleled advantages in terms of operating cost and energy efficiency when compared to the conventional solvent process. Moreover, it can pave a path to battery miniaturization as the absence of solvent elevates the maximum threshold of active mass loading, allowing the fabrication of higher mass-loading electrodes[14–18]. The recent progress in dry LIB electrode technology involves dry-pressing a mixture of LiFePO$_4$ (LFP) active material powder and holey graphene to form a freestanding composite electrode. The use of holey graphene results in a binderless electrode configuration with a rate capability comparable to that of conventional LFP electrodes[19]. However, the prepared electrode has proven unsuitable for electrode roll-to-roll fabrication, as it requires exceedingly high pressure (20–500 MPa) for its fabrication and is easily fractured when bent. Another method involves electrostatically spray-coating the electrode material onto the Al current collector, followed by hot roll press compaction of a dry LiCoO$_2$ (LCO) electrode. However, the scalability of dry spray-coating remains in question, and the additional coating step also complicates the manufacturing process[20,21]. These previous reports on the dry LIB electrode process have mainly focused on improving performance by changing the coating process or the binder, but alternative ways such as employing new conductive agents or current collectors have been rarely explored in addressing the core challenges of solvent-free electrode fabrication, which include weak cohesive strength, low deformability, high cell polarization, and low rate capability.

[1]Department of Chemical and Biomolecular Engineering, Yonsei University, Seodaemun-gu, Seoul 03722, Republic of Korea. ✉e-mail: lutts@yonsei.ac.kr

Carbon nanotubes (CNTs) are among the most avidly studied and utilized materials for multipurpose LIB electrode fabrication owing to their remarkable electronic conductivity, mechanical strength, resistance to chemical degradation, etc[22,23]. To the best of our knowledge, there is limited information on the use of dry powdered CNTs and a polymeric binder to directly press-coat electrode material onto a current collector via a completely solvent-free approach. Therefore, the dry press-coating capability of the multiwalled carbon nanotube (MWNT)-polyvinylidene fluoride (PVDF) composite powder was evaluated for the first time by measuring its adhesive and cohesive strength upon pressing. Additionally, etched Al foil was selected as a new current collector (Supplementary Fig. 1) to enhance adhesion by inducing an anchoring effect on the submicron pores of the foil surface. According to reports, the interfacial contact between the electrode film and the substrate surface was improved by the larger contact area of the $Al_2O_3$ passive layer[24–26]. As shown in the X-ray photoelectron spectroscopy (XPS) results (Supplementary Fig. 2), the Al $2p$ photoelectron spectra indicated a higher amount of the $Al_2O_3$ layer on the etched Al foil than on the normal Al foil.

In this study, we develop a novel method for the fabrication of a solvent-free $LiNi_{0.7}Co_{0.1}Mn_{0.2}O_2$ (NCM712) electrode, namely, a dry press-coated electrode (DPCE), via the facile one-step hot-pressing of premixed NCM712, MWNTs, and a dry PVDF powder mixture onto etched Al foil (Fig. 1 and Supplementary Fig. 3). Additionally, the influences of the MWNT and binder content on the electrode structure and electrochemical performance are evaluated to find the optimal electrode composition. The optimally composed DPCE is then compared with conventional slurry-coated electrodes (SCEs) on various aspects, such as morphology and electrochemical performance. Furthermore, high loading DPCE pouch cells are fabricated to manifest an excellent three-dimensional (3D) conductive network.

## Results

### The adhesive and cohesive properties of the MWNT and PVDF composite against the etched Al foil

To explore the dry press-coating capability of the MWNT and PVDF composite powder, the dry MWNT-PVDF composite (d-MP) with a weight ratio of 1:1 was mechanically pressed onto an etched Al foil by both cold-pressing (25 °C) and hot-pressing (180 °C) (to manifest the effect of pressing temperature) at an equal pressure of 10 MPa for 30 s. The temperature of the press was set above the melting point of PVDF (177 °C) to initiate thermal activation of the binder and produce a bonding effect[27]. For comparison, the dry Super P-PVDF composite (d-SP) with the same weight ratio was also pressed under the same conditions, and the results were juxtaposed. Additionally, the as-prepared electrodes were folded to test their brittleness (Fig. 2a-d).

Notably, the results showed that the dry MWNT-PVDF composite, which was hot-pressed onto the etched Al foil (Fig. 2d) had the most uniform coating layer when compared to others and showed no trace of crevice even after folding. When the comparative experiment was performed by substituting the current collector with normal Al foil (Supplementary Fig. 4), the adhesion of the dry MWNT-PVDF composite against the etched Al foil was far more noticeable, indicating a superior solid-solid adhesion at the interlayer. Moreover, the scanning electron microscopy (SEM) images of the interface between the MWNT-PVDF scaffold and the substrate (etched Al foil) revealed a web-like MWNT-PVDF composite network anchored firmly onto the corrugated surface of the etched Al foil current collector (Supplementary Fig. 4).

Additionally, the pull-off test was conducted to investigate the cohesive property of the MWNT-PVDF composite[28,29]. In our case, it was customized to measure the force required to completely separate the sandwiched dry electrode made by hot-pressing different dry powder samples between two etched Al foils. Thus, three different sandwiched dry electrodes were prepared using MWNT-PVDF (d-MP), PVDF, and Super P-PVDF (d-SP) powder, and their respective tensile strengths were comparatively measured by stripping the electrodes in the opposite z-axis directions (Fig. 2e). Remarkably, the tensile strength of d-MP sandwiched dry electrode was significantly higher than that of the d-SP and PVDF sandwiched dry electrodes, indicating the presence of superior cohesion and synergistic interaction between MWNT and PVDF (Fig. 2f). This outstanding tensile strength of the d-MP electrode can be attributed to the tightly intertwined MWNT-PVDF network, which is capable of inducing the nano hook-and-loop fastening effect within the electrode body and concatenating the entire structure[30]. Moreover, the MWNT-PVDF composite exhibited favorable adhesive and cohesive properties with the etched Al foil; thus, it was chosen as the main ingredient for DPCE fabrication in this work.

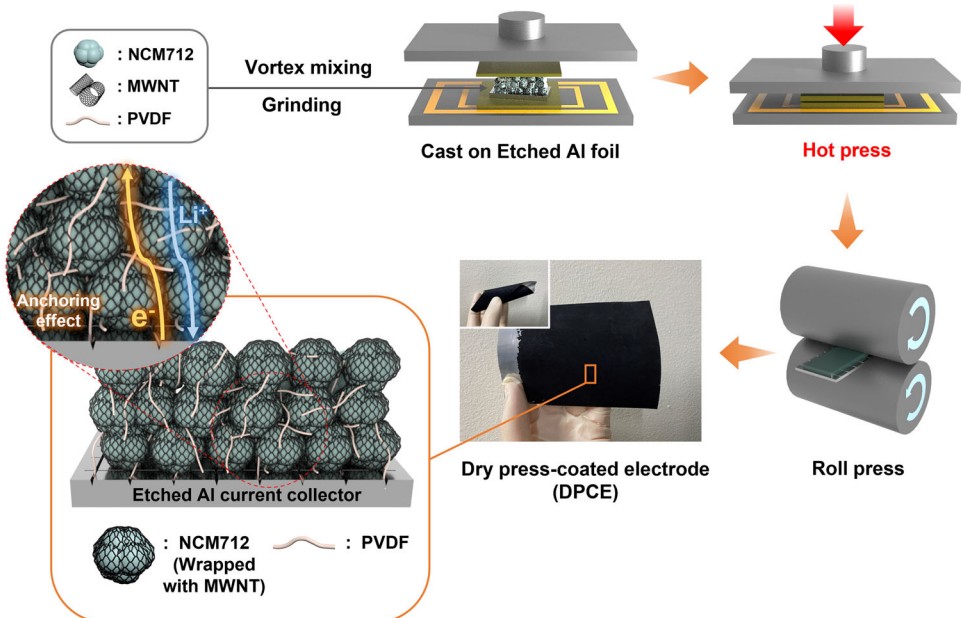

**Fig. 1 | Fabrication of the dry press-coated electrode (DPCE).** Schematic illustration of the fabrication procedure and structural design of the DPCE.

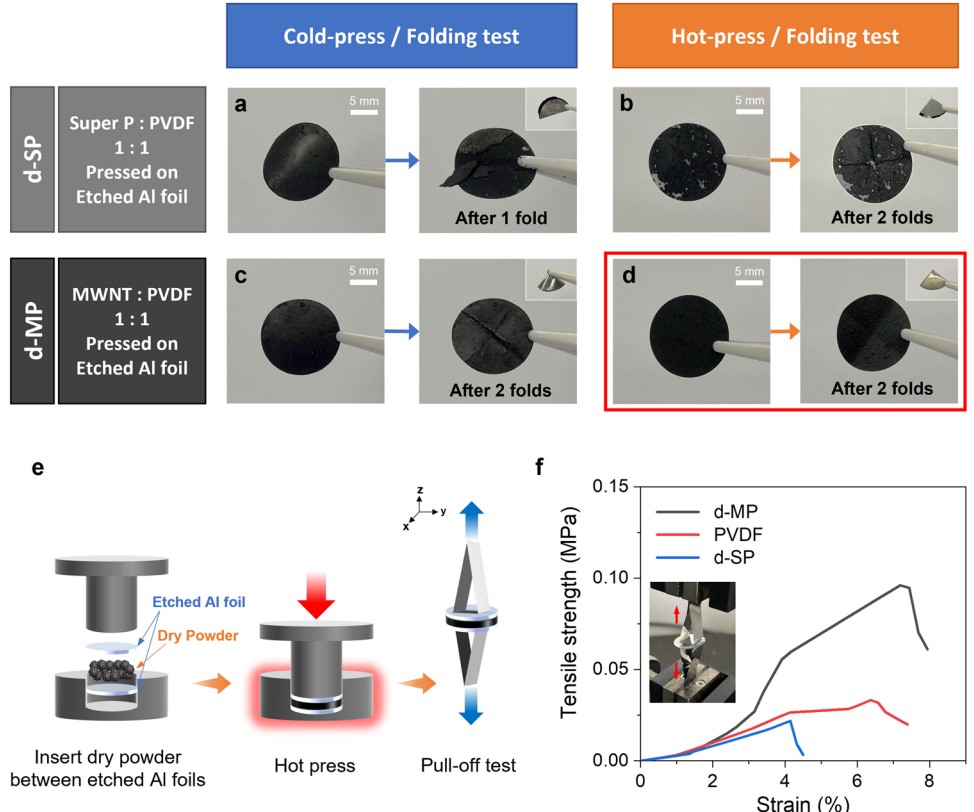

**Fig. 2 | Comparison of the dry press-coating capability of the conductive scaffolds. a–d** Folding test of Super P:PVDF–1:1 dry electrode (d-SP) after cold/hot press **a**, **b**. Folding test of MWNT:PVDF–1:1 dry electrode (d-MP) after cold/hot press **c**, **d**. **e** Schematic illustration of the tensile strength measurement setup. **f** Comparison of the stress-strain curves of d-MP, PVDF, and d-SP sandwiched dry electrode; diameter of the disc is 15 mm.

## Effects of the MWNT and PVDF binder contents on the mechanical and electrochemical properties of the DPCE

To determine the applicability of MWNT-PVDF composite in DPCE fabrication, the effects of MWNT and PVDF content on the mechanical and electrochemical properties of the DPCE were investigated. First, the mechanical properties of the DPCE samples with varying MWNT and PVDF ratios (see Methods section for details) were evaluated using a 180° peel test. According to the peel test results (Supplementary Fig. 5a), the DPCE 0515 (4.1 N cm$^{-1}$) and DPCE 1010 (4.0 N cm$^{-1}$) showed higher average adhesion forces than DPCE 1505 (3.2 N cm$^{-1}$), supporting the logic that an increase in binder content enhances the adhesive strength with the current collector. As displayed in the stripped tape images (Supplementary Fig. 5b), the severity of the electrode detachment decreased as the binder content increased. However, there was no significant decrease in the adhesive strength of the electrode as the binder content decreased from 15 to 10 wt%, indicating that the corresponding increase in MWNT content (5 to 10 wt%) created a more concrete and supportive scaffold that enhanced the attachment of the electrode layer onto the etched Al foil surface.

Notably, due to the facile and strong adhesive properties of dry press-coating technology, the fabrication of double-sided electrodes can be expedited through a simple two-step process that involves continuous dry press-coating on both sides of the etched Al foil (Supplementary Fig. 6). To verify the rigidity of the electrode, the as-prepared electrode was twisted several times, and then a cross-sectional SEM image was obtained to analyze the integrity of the electrode material. The double-sided DPCE remained intact even after several rough twisting cycles, and the electrode material incurred negligible damage and exfoliation, demonstrating the applicability of the dry press-coating process in multilayered pouch cell fabrication.

The dispersity of the electrode materials is also crucial in determining the mechanical strength and electrochemical performance of the electrode[31,32]. Obtaining homogeneous dispersion of untreated MWNTs has been a major challenge in solvent-based processes owing to agglomeration caused by the high aspect ratio and strong van der Waals interactions of the nanotubes[33,34]. Fortunately, such issues can be circumvented in the dry press-coating process. As observed from the elemental mapping images of the DPCE with different MWNT and PVDF contents (Supplementary Fig. 7 and Supplementary Fig. 8), the material dispersity of the electrode improved as the MWNT content increased relative to the PVDF content. Consequently, DPCE 1505 (with the highest MWNT content) displayed a much more even material distribution than DPCE 1010 and DPCE 0515, which contained interspersed binder aggregates (and voids) that segregated active materials. These inactive parts can act as insulators and hinder the transport of electrons within the electrode. Therefore, it is important to note that the formation of unwanted aggregates can be mitigated by controlling the ratio of the MWNT and PVDF content in the electrode architecture.

Finally, the optimization of the DPCE composition was performed by comparing the electrochemical performances of the three different compositions of DPCEs using a half-cell test (Supplementary Fig. 9). As shown in Supplementary Fig. 9a, b, DPCE 1505 outperformed the other two electrodes in terms of discharge rate capability and long-term cycling performance, exhibiting the highest reversible discharge capacity (149.7 mAh g$^{-1}$) at 5.0 C and capacity retention (77.7%) after 250 cycles at 0.5 C. The electrochemical impedance spectroscopy (EIS) experiments were also performed to analyze the electron/ionic transportation behaviors of the electrodes. As shown in the Nyquist plots (Supplementary Fig. 9c), DPCE 1505 showed much smaller ohmic resistance ($R_o$) (2.2 Ω) and charge transfer resistance ($R_{ct}$) (54 Ω) compared to DPCE 1010 (2.4 and 64 Ω) and DPCE 0515 (4.2 and 88 Ω).

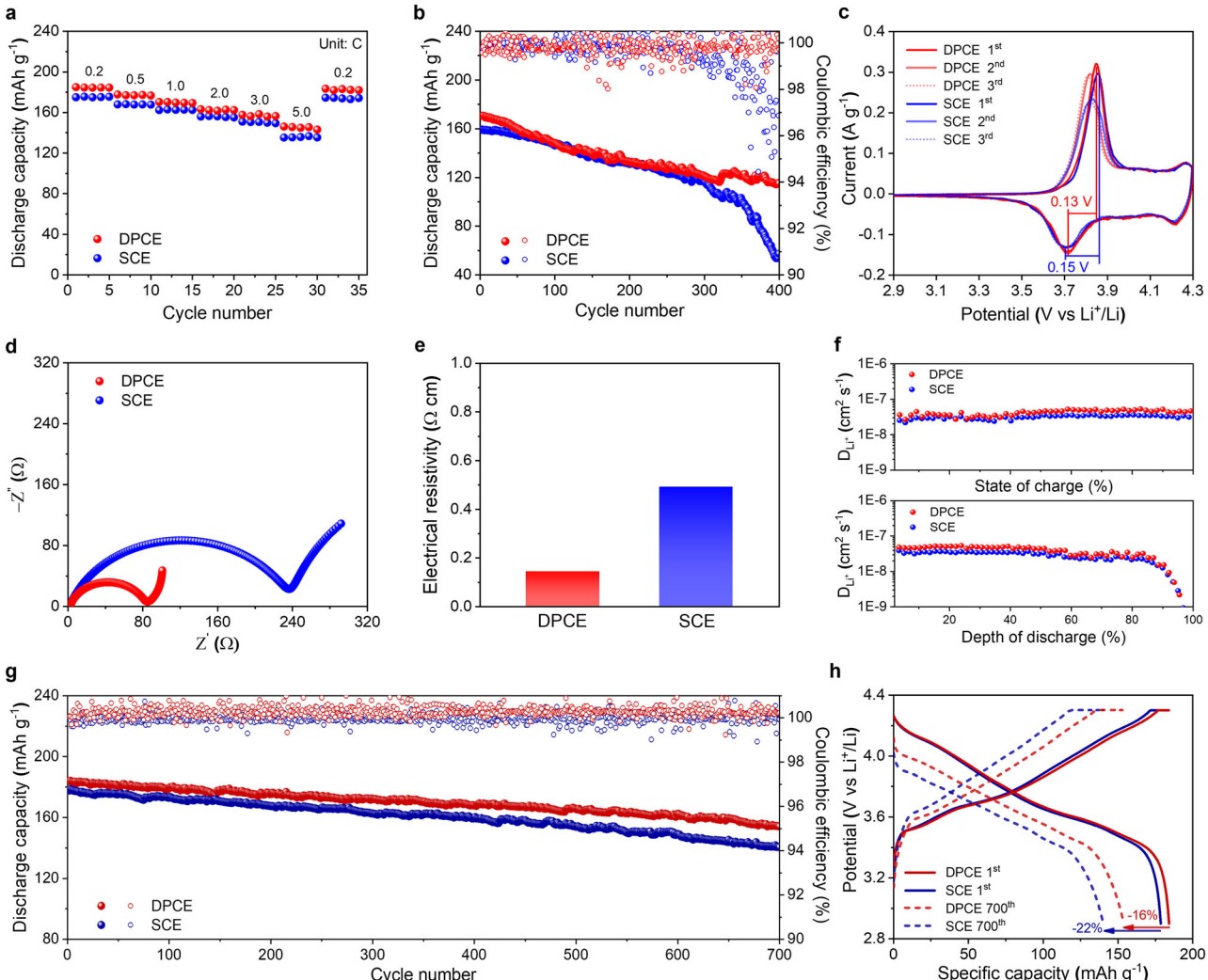

**Fig. 3 | The electrochemical performance comparison of the SCE and DPCE.** **a** Rate capability. **b** Half-cell cycling performance at a charge/discharge current density of 1.0/1.0 C. **c** CV profile of both electrodes at a scan rate of 0.2 mV s$^{-1}$ between 2.9 and 4.3 V. **d** Fitted Nyquist plots. **e** Surface electrical resistivity of the

DPCE vs. SCE. **f** Calculated diffusion coefficient as a function of SOC and DOD. **g** Full-cell cycling performance at a charge/discharge current density of 0.5/0.5 C. **h** Charge/discharge voltage profile comparison (full-cell).

Additionally, the charge-discharge voltage profiles of the three DPCEs during long-term cycling (Supplementary Fig. 9d–f) manifest the significantly small capacity decay and overpotential increase in DPCE 1505 over other electrodes, confirming the significance of MWNT and PVDF ratio on the electrochemical performance of DPCEs. Thus, based on these results, DPCE 1505 (NCM712/MWNT/PVDF – 80/15/5) was chosen as the representative cathode. Furthermore, using this optimal composition, other active materials, such as NCM622, LCO, and LFP, were also tested in fabricating DPCEs, and they all exhibited excellent rate capabilities (Supplementary Fig. 10), verifying the versatility of the dry press-coating process.

**Morphological analysis and electrochemical performance comparisons of the DPCEs and conventional SCEs**
The morphologies of the DPCEs and SCEs (with equal compositions) were first examined using the top and cross SEM images (Supplementary Fig. 11). It was found that the DPCE architecture had a much denser and continuous conductive network with lesser voids when compared to SCE. Presumably, this is due to the sheet-to-point contact mode present in DPCEs as opposed to the line-to-point contact mode prevalent in SCEs[35]. The PVDF binder facilitates the interweaving of the dispersed MWNT subunits and creates a web structure that tightly

secures the active material within its matrix[36,37]. This contact mode is more effective, especially in high-loading electrodes because an increase in the number of composite sheet layers enhances the structural rigidity of the electrode body and allows the mechanical stability to be maintained as the electrode thickness increases[38]. Additionally, a contact angle measurement was used to investigate the electrolyte wettability behavior (Supplementary Fig. 12), and the result showed that the DPCE had greater electrolyte wettability than the SCE with a smaller initial contact angle (15 vs. 24°) and a faster penetration rate. The swelling resistance of the electrodes was also tested (Supplementary Fig. 13). After being immersed in the electrolyte solution and ultrasonicated for 3 min, the DPCE retained its original structure and the electrolyte solution remained clear, while the SCE incurred delamination due to surface exfoliation, and the electrolyte solution became turbid. Consequently, all these results indicate the presence of a robust 3D conductive framework in the DPCE as well as abundant nanochannels for rapid ion transport.

The electrochemical performance comparisons of the DPCE and SCE are displayed in Fig. 3. Notably, the DPCE exhibited better rate capability than the SCE at all current densities as shown in Fig. 3a. Moreover, a long-term cycling test was performed using a half-cell at 1.0 C with a mass loading of both electrodes at 8–9 mg cm$^{-2}$ (Fig. 3b).

It is noteworthy that the DPCE demonstrated much better cycling stability than the SCE with an initial capacity of 170 mAh g$^{-1}$ (1.0 C) and capacity retention of 67% after 400 cycles along with stable coulombic efficiency. In contrast, the SCE delivered an initial capacity of 159 mAh g$^{-1}$ (1.0 C) with a much lower capacity retention of 35% after 400 cycles and unstable coulombic efficiency after 300 cycles. This result can be explained by postmortem analysis of the coin cell after cycling, wherein the DPCE retained its original structure with negligible cracks, while the SCE showed detachments with noticeable voids around the active materials after cycling (Supplementary Fig. 14).

Additionally, cyclic voltammetry (CV) was performed at a scan rate of 0.2 mV s$^{-1}$ (2.9–4.3 V) to verify the underlying mechanism that allows the DPCE to outperform the SCE. The initial 3 curves are shown in Fig. 3c along with a pair of major redox couples in the potential range of 3.6–3.9 V for both the DPCE and SCE, corresponding to the Ni$^{2+}$/Ni$^{4+}$ redox process[39]. According to the CV results, the DPCE had a higher peak current and smaller peak voltage difference than the SCE, which exemplifies that the DPCE has faster electrochemical activity and better reversible capacity for lithium-ion insertion/extraction. The Nyquist plot of the DPCE in Fig. 3d demonstrates a much lower impedance in the high-medium frequency region, corresponding to R$_{ct}$ than that of the SCE (57 vs. 196 Ω). This was further verified by comparing the surface electrical resistivity of the two electrodes (Fig. 3e). The galvanostatic intermittent titration technique (GITT) was used to better understand the ion transportation behavior of the DPCE (Supplementary Fig. 15). Figure 3f shows the GITT-based calculated ionic diffusion coefficient results as a function of the state-of-charge (SOC) and depth-of-discharge (DOD). The DPCE exhibited an overall higher diffusion coefficient (4.22 × 10$^{-8}$ cm$^2$ s$^{-1}$) than the SCE (3.13 × 10$^{-8}$ cm$^2$ s$^{-1}$), indicating that the DPCE had a faster ionic transportation despite its greater thickness.

Furthermore, a full-cell test was conducted using a graphite anode as the counter electrode (graphite/Super P/CMC – 90/5/5, made via slurry coating process). Both the DPCE and SCE full-cells were assembled with a mass loading of 10.5–10.7 mg cm$^{-2}$ and an N/P ratio of 1.13. Long-term cycling tests were performed in the voltage range of 2.9–4.3 V at 0.5/0.5 C (charge/discharge) after one precycle at 0.1/0.1 C. The results in Fig. 3g, h show that the DPCE delivered a higher initial capacity of 184 mAh g$^{-1}$ with a capacity retention of 84% after 700 cycles. In contrast, the SCE exhibited a lower initial capacity of 177 mAh g$^{-1}$ with a lower capacity retention of 78% after 700 cycles. The cycling performance at a higher C-rate (1.0 C) also demonstrated better cycling stability and greater lithium intercalation/deintercalation capability of the DPCE compared to SCE (Supplementary Fig. 16).

To elucidate the inner frameworks of the two electrodes, the DPCE and SCE were comparatively analyzed by microcomputed tomography (micro-CT) (Fig. 4a). The obtained 3D images could be segmented into two phases, an active material phase (NCM712, green) and a carbon phase (Carbon, red), because of their sufficient difference in absorption contrast[40]. As presented in Fig. 4a, the DPCE exhibited thick and compact packing of the active materials in its bicontinuous carbon structure, confirming the presence of a robust conductive framework. However, the SCE showed multiple cracks and a substantially smaller loading of active materials owing to the inherent limitations of the slurry-coating process. The cycled DPCE and SCE cells were also analyzed by inductively coupled plasma mass spectrometry (ICP–MS) (Fig. 4b). The result showed that there was fewer amount of transition metals (Ni, Mn, and Co) deposited on the Li-metal anode when coupled with the DPCE than when coupled with the SCE. In line with the ICP–MS results, the time-of-flight secondary ion mass spectroscopy (TOF–SIMS) mapping images of the DPCE (versus SCE) surface in Fig. 4c, reveals a significantly smaller NiF$_2^+$ formation compared to SCE. This was attributed to the side reactions between NCM712 and electrolytes[41–44]. To further validate these results, the XPS F 1 $s$ spectra (Fig. 4d, e) of both DPCE and SCE were examined after the cycling test, which demonstrated much lower emergence of NiF$_2$ and Li$_x$PO$_y$F$_z$ peaks on the DPCE surface than on the SCE surface (≈684.5 eV and ≈686.6 eV, respectively)[45,46].

## High mass loading capability of the dry press-coating process

Another advantageous feature of the dry press-coating method is the fabrication of high mass-loading electrodes. Using the DPCE 1505 composition, cathodes with varying areal mass loadings of 22 mg cm$^{-2}$ (98 μm), 25 mg cm$^{-2}$ (123 μm), 28 mg cm$^{-2}$ (176 μm), and 30 mg cm$^{-2}$ (221 μm) were fabricated using the dry press-coating method. The cross-sectional SEM images of the as-prepared high-loading DPCEs (HL–DPCEs) are arranged in Fig. 5a in the order of increasing areal mass loading (with their corresponding thicknesses). The overall morphologies of the electrodes exhibited a compact and uniform packing layer with no visible voids or cracks within the structure. As shown in the voltage profile plots in Fig. 5b, the areal capacities of all the electrodes increased as the areal mass loading increased. The cycling performances of the Li-metal half-cells with varied areal mass loadings of DPCE were tested at a high charge/discharge current density of 1.0/1.0 C (Fig. 5c). The results show that all the electrodes exhibited cyclic stability throughout the measuring cycles, confirming the excellent charge transfer kinetics of the thick electrode configuration.

For comparison with the high loading SCEs (HL–SCEs), several attempts were made to elevate the SCE loading (to a comparable level) via casting a highly viscous slurry onto the Al foil, but all the as-prepared electrodes incurred severe delamination due to structural collapse caused by solvent evaporation. In contrast, when the current collector was replaced by the etched Al foil, the damage was significantly lower, and there was no delamination after drying (Supplementary Fig. 17); thus, this electrode was used for comparison. As displayed in the 180° peel test results (Fig. 5d), the HL–DPCE (98 μm thick) exhibited a more outstanding adhesion strength than the HL–SCE (80 μm thick) (4.7 vs. 0.5 N cm$^{-1}$), exemplifying that DPCEs have better compatibility to a high loading electrode architecture than SCEs. The cycling performances of the HL–DPCE (22 mg cm$^{-2}$) and the HL–SCE (16 mg cm$^{-2}$) were also compared at a charge/discharge current density of 1.0/1.0 C (Fig. 5e). Despite having 40% higher areal mass loading, the HL–DPCE showed much greater cycling stability than the HL–SCE with a capacity retention of 87% and stable coulombic efficiency after 80 cycles. In contrast, the HL–SCE showed a rapid decay in capacity after 50 cycles owing to its sluggish charge transfer kinetics and dissolution of the electrode materials in the electrolyte solution during the cycling process[42,47].

Furthermore, a Li-metal pouch cell (40 × 35 mm$^2$ in size) was fabricated using the DPCE coupled with the Li-metal anode (200 μm) (Supplementary Fig. 18), and its performance was evaluated to explore the practicability of the HL–DPCE cells. As demonstrated in the voltage profiles of the fabricated DPCE pouch cell (31 mg cm$^{-2}$), an initial discharge capacity of 187 mAh g$^{-1}$, which is a near-theoretical capacity of NCM712 (190 mAh g$^{-1}$) at 0.1 C, could be achieved (Fig. 6a). The long-term cycling test was also conducted using the same pouch cell at a high charge/discharge current density of 0.5/0.5 C, and the result showed a stable cycling performance with a capacity retention of 85% after 100 cycles, indicating a superb rate capability and cyclability of the DPCE (Fig. 6b). To test the high loading capability of the DPCE, Li-metal pouch cells with various areal mass loadings were fabricated (Fig. 6c). The cycling test showed stable cycling performance, emphasizing the applicability of the DPCE at extremely high loading conditions (Fig. 6d). Unsurprisingly for such high loading electrodes, the discharge capacity at a higher C-rate (0.5 C corresponding to 6.76 mA cm$^{-2}$) was relatively low (Supplementary Fig. 19), presumably due to a combined effects of high porosity and high current density. Future studies will be dedicated to improving electron percolation pathways and material contact by enhancing the homogeneity of particle packing and micro-patterning of current collector.

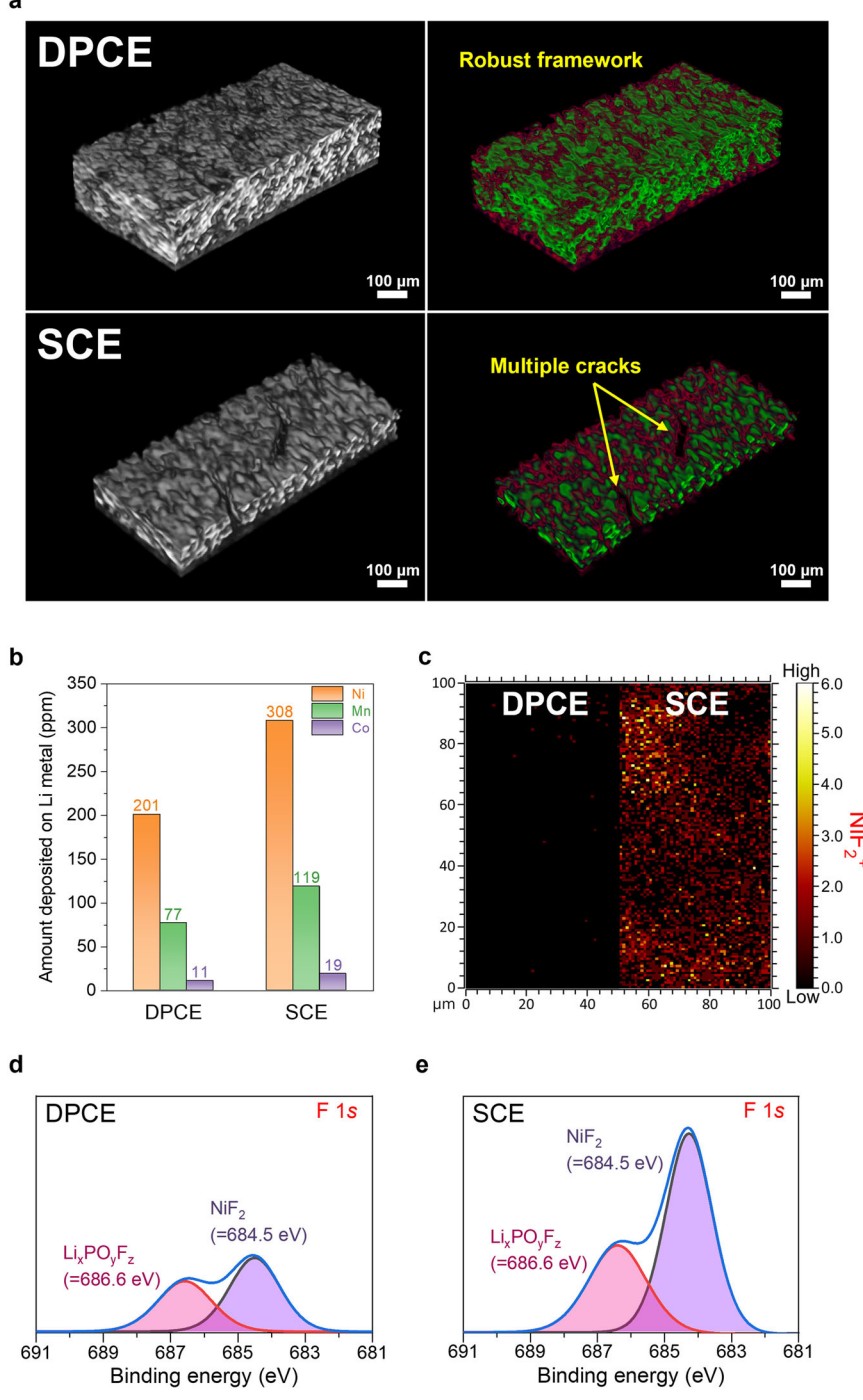

**Fig. 4 | Morphologies and postmortem analysis of the cycled DPCE (versus SCE). a** Micro-CT images of the DPCE and SCE. The active material (NCM712) phase is labeled green, and the carbon phase is labeled red. **b** Amount of metallic Ni, Mn, and Co deposited on the Li-metal anodes (ICP–MS). **c** TOF–SIMS mapping images of the NiF$_2^+$ byproducts formed on the surface of the cathodes. **d**, **e** XPS F 1 s spectra of the cycled DPCE **d** and SCE **e**.

Nonetheless, this high loading design has enabled the fabrication of a high energy density Li-metal pouch cell (equipped with 17.6 mAh cm$^{-2}$ DPCE), and the specific energy and volumetric energy density, calculated based on the entire cell weight were up to 360 Wh kg$^{-1}$ and 701 Wh L$^{-1}$, respectively, in the initial cycle at 0.1 C (i.e., 1.7 mA cm$^{-2}$) (Supplementary Table 1). It is worth noting that the high loading capability of the DPCE pouch cell far exceeded that of previously reported solvent-free electrodes in terms of both areal capacity and mass loading (Fig. 6e and Supplementary Table 2), even outperforming other high-loading cathodes fabricated using different methods (Fig. 6f and Supplementary Table 3).

## Discussion

In this work, the dry press-coating process, a novel dry process for LIB electrode fabrication, was successfully demonstrated using a MWNT-PVDF composite as the active material host and an etched Al foil as a current collector. Notably, the DPCEs fabricated using this process exhibited both strong adhesion and cohesion properties because of their bicontinuous intertwined structure, which is capable of anchoring onto the submicron pores of the etched Al foil and enabling seamless compaction of active materials within its web-like network. It was further revealed that these unique properties of DPCE allowed the fabrication of high loading electrode, and as a

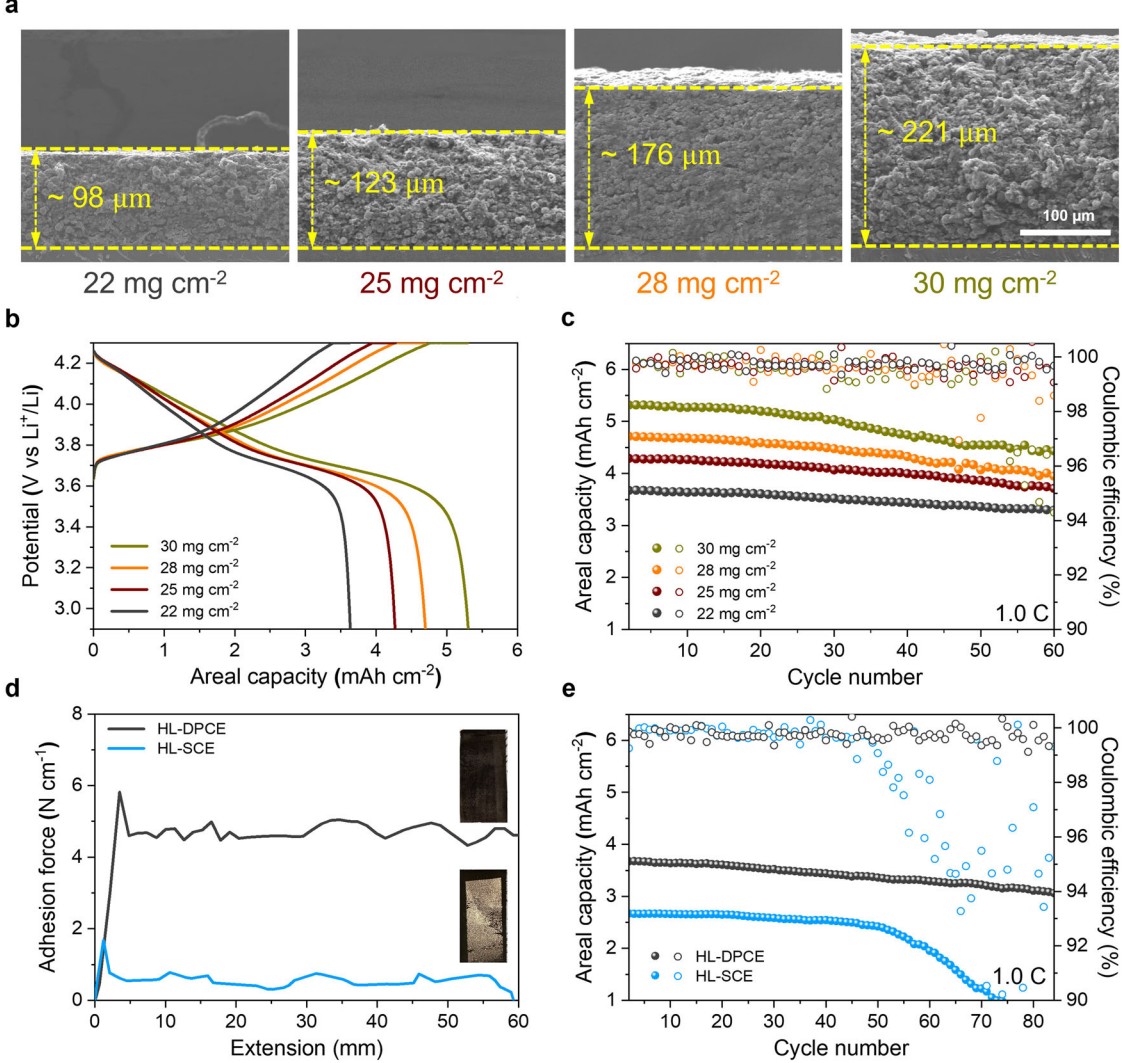

**Fig. 5 | The high mass loading DPCEs (coin cells). a** Cross-section SEM images of the DPCE with varying mass loadings and their corresponding thicknesses. **b** Charge/discharge voltage profiles (in terms of areal capacities). **c** Cycling performance of the DPCE with different areal mass loadings at a charge/discharge current density of 1.0/1.0 C at a voltage range of 2.9–4.3 V. **d** Adhesion force of the high-loading DPCE (HL−DPCE) and high-loading SCE (HL−SCE), which was estimated using 180° peel-off test. The insets show the digital photos of the electrodes after the peel-off test. **e** Comparison of the cycling performance of the HL−DPCE and HL−SCE at a charge/discharge current density of 1.0/1.0 C at a voltage range of 2.9–4.3 V.

proof of concept, the Li-metal pouch cell was assembled using the DPCE with an areal mass loading of 100 mg cm$^{-2}$ (corresponding to an areal capacity of 17.6 mAh cm$^{-2}$), which demonstrated a specific energy and volumetric energy density of 360 Wh kg$^{-1}$ and 701 Wh L$^{-1}$ (based on the entire cell weight), respectively, confirming the practicability of DPCEs. Furthermore, the excellent compatibility of DPCEs with other active materials makes the dry press-coating process a promising solution for the scalable manufacturing of dry LIB electrodes.

## Methods
### DPCE fabrication
NCM712, MWNTs, and PVDF binder were first premixed using a mortar. The amount of active material was fixed at 80 wt%, while the amount of MWNTs and PVDF binder were varied with the overall weight ratio compositions of 80/15/5, 80/10/10, and 80/5/15, for the fabrication of DPCE 1505, DPCE 1010, and DPCE 0515, respectively (number refers to the weight ratio of MWNT and PVDF). All powders were dried in a vacuum oven at 80 °C before use. The premixed powder was then transferred to a vial and mixed using a vortex agitator for 1 min. Vortex mixer was used instead of ball milling to prevent damage to the MWNTs and preserve the intertwined network structure. Subsequently, a piece of etched Al foil was placed on a steel plate base, and the electrode mixture was spread evenly onto the foil. Before sealing with the upper steel plate, a sheet of stainless foil was placed over the powder to prevent unwanted material adhesion to the upper steel plate. The sealed plates with electrode inside were then transferred to a heat press (that had been preheated to 180 °C), and a load of 10 MPa was applied for 30 s. After pressing, the as-prepared DPCE was roll-pressed at room temperature.

### SCE fabrication
SCEs were fabricated by mixing NCM712, Super P, and PVDF in NMP with a weight ratio of 80/15/5. The slurry was mixed using a Thinky mixer (Thinky USA) at 2000 rpm for 20 min and then was coated onto etched Al foil with a film coater. Subsequently, the as-prepared electrode was vacuum dried at 110 °C for 12 h and pressed through a roll press.

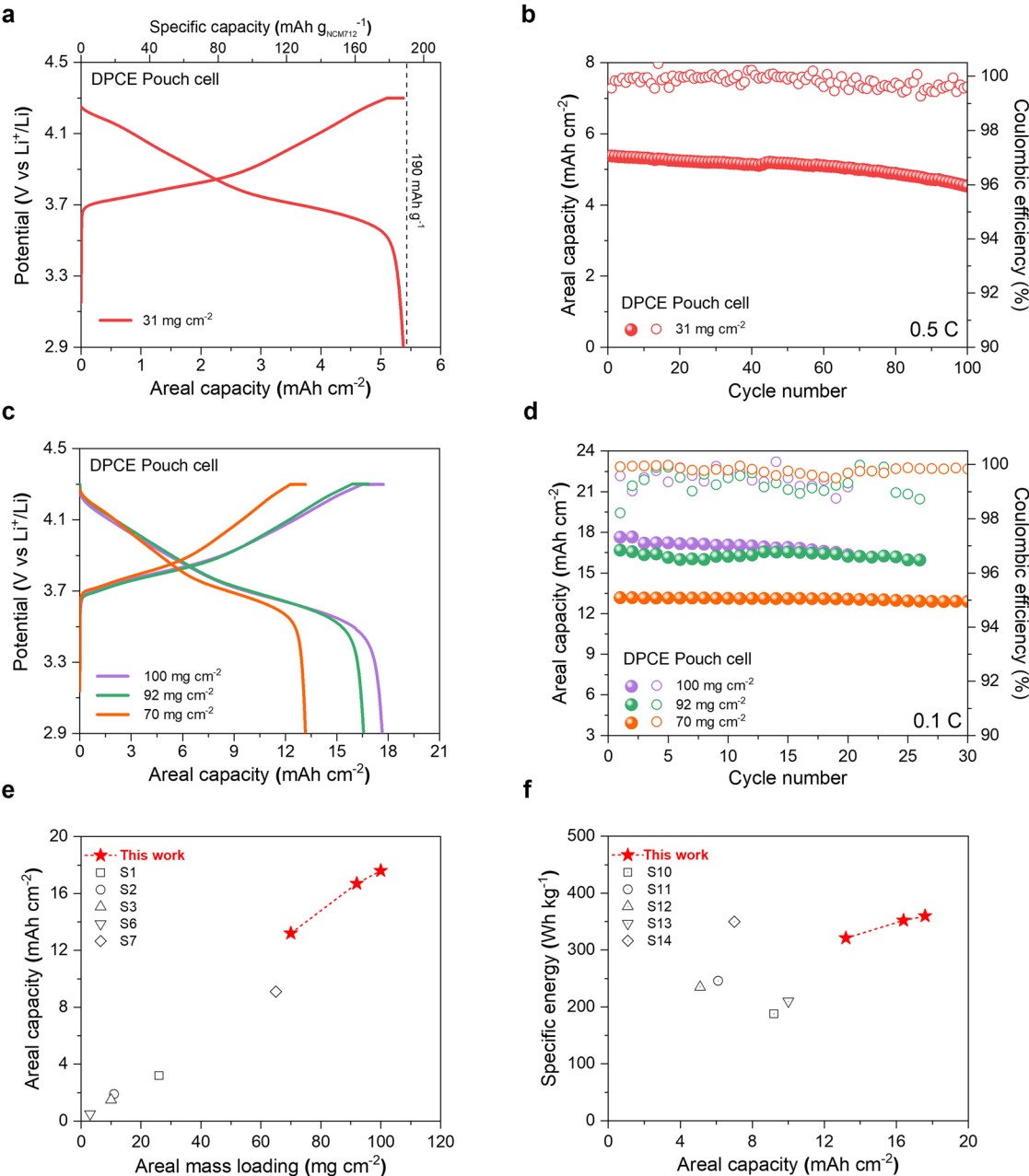

**Fig. 6 | The high mass loading DPCEs (Pouch-cells). a** Charge/discharge voltage profile of the Li-metal DPCE pouch cell (31 mg cm⁻²) at a voltage range of 2.9–4.3 V and charge/discharge current density of 0.1/0.1 C (dashed line represents the theoretical capacity of NCM712). **b** Cycling performance of the Li-metal DPCE pouch cell (31 mg cm⁻²) at a charge/discharge current density of 0.5/0.5 C at a voltage range of 2.9–4.3 V. **c** Charge/discharge voltage profiles of the Li-metal DPCE pouch cells in terms of areal mass loadings at a voltage range of 2.9–4.3 V and charge/

discharge current density of 0.05/0.1 C. **d** Cycling performance of the Li-metal DPCE pouch cells with various areal mass loadings. **e** Areal capacities of cells as a function of areal mass loading (DPCEs versus previously reported solvent-free cathodes). **f** Specific energies (calculated based on the entire cell weight) of cells as a function of areal capacity (DPCEs versus previously reported high mass loading cathodes).

## Materials and physiochemical characterization

NCM712 (L&F Co.) was used as the active material, MWNTs (20 nm in diameter, JEIO Co., Korea) were used as conductive agents, and PVDF was used as the binder. The etched aluminum foil (JCC Co., Korea; 20 μm, > 99.7% purity; made by chemical etching) was used as the current collector. XPS data were acquired using the K-Alpha system (Thermo VG, U.K.) to analyze the chemical composition of the current collectors and to detect the by-products occurring from the side reactions between electrolyte and electrode. The surface and cross-sectional morphologies of the DPCEs and SCEs were analyzed by SEM (7610F-Plus, JEOL Ltd.) and energy-

dispersive X-ray spectroscopy (EDS). Cross-sectional images were obtained from crosscuts using an argon ion beam (Cross Section Polisher (CP) IB-19510, JEOL). The adhesion strength and tensile strength of the DPCEs were investigated using a 180° peel test and pull-off test using a universal testing machine (Lloyd Instruments) at a peel speed of 250 mm min⁻¹. The electrical resistivity of the electrodes was measured using an electrode measurement system (RM2610, Hioki). For the postmortem analysis, the inductively coupled plasma-mass spectroscopy (ICP–MS, Agilent 7900) was used to analyze the amount of transition metals (Ni, Mn, and Co) deposited on the Li-metal anode. The surface analysis of the cycled

cathode surface was conducted using time-of-flight secondary ion mass-spectroscopy (TOF–SIMS, ION TOF) with a $Bi_3^{2+}$ gun (30 keV, 1 pA).

## Electrochemical characterization

The prepared DPCEs and SCEs were punched into discs with diameters of 14 mm, and 2032-type coin cells were used to test the electrochemical performance. A polyethylene (PE) separator and a liquid electrolyte solution consisting of 1 M $LiPF_6$ in ethylene carbonate/diethyl carbonate (EC/DEC 1/1 (w/w)) with 10 wt% fluoroethylene carbonate (FEC) were used throughout the test. For the half-cell tests, a Li-metal anode (300 µm, Honjo Metal) was used as the counter and reference electrode, and full-cell tests were conducted using a graphite anode (graphite/Super P/CMC – 90/5/5, made via slurry coating process) as the counter electrode. The assembly was carried out in an argon-filled glove box. The cell performance and GITT measurements were conducted using a battery cycler (WBCS 3000, WonATech) in a chamber (set as 25 °C), under the constant current-constant voltage (CC–CV) procedure in the voltage range of 2.9–4.3 V vs. Li/Li⁺. One precycle at 0.1 C was performed as a formation step for all cells before the tests. The rate capability test was conducted at a fixed charging rate of 0.2 C with a varying discharging rate from 0.2–5.0 C, and a return to 0.2 C to ensure capacity recovery. The cycle test was measured under various charging/discharging conditions. CV and EIS measurements were performed using an electrochemical workstation (ZIVE MP1, WonATech) with a scan rate of 0.2 mV s⁻¹ for CV and a frequency range between $10^{-2}$ Hz–$10^5$ Hz with a 10 mV AC signal. The lithium metal pouch cell was fabricated using an Al pouch film as a package and lithium metal (200 µm) as an anode in a dry room with a dew point of −50 °C. The electrochemical performance of the pouch cell was evaluated after the initial formation for 2 cycles at 0.05 C under a fixed pressure set at 500 kPa.

## Calculation of the specific energy and volumetric energy density

The specific energies and volumetric energy densities of the pouch cell were calculated based on the experimentally measured weight and volume of the cell including the packaging film. Calculation details for the specific energies and specific densities are described in Supplementary Table 1.

## Data availability

The authors declare that the main data supporting the findings of this study are available within the paper and its Supplementary information. Extra data are available on reasonable request from the corresponding author.

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

## Acknowledgements

This research was supported by the National Research Foundation of Korea (NRF) funded by Ministry of Science and ICT (2018M3D1A1058624 and 2019R1A2C3010479).

## Author contributions

M.R. and J.H.P. designed this work. M.R. performed the experimental characterization and electrochemical tests. M.R. and Y.K.H. participated in the fabrication of dry press-coated electrode (DPCE) pouch cell. J.H.P. and S.Y.L. supervised the overall project. M.R. analyzed the results and wrote the manuscript. All the authors discussed the results presented in the manuscript.

## Competing interests

The authors declare no competing interests.
