## [Peer Review File · Nature Communications]

Ultrahigh loading dry-process for solvent-free lithium-ion battery electrode fabricationReviewers' comments:

Reviewer #1 (Remarks to the Author):

This work reports a dry press-coating technique to fabricate the high loading electrode for lithium ion batteries. The authors use MWNT / PVDF composite scaffold as the active material host and an etched Al foil as a current collector. With the mixing, casting, hot press, and roll process, the thick electrode with strong internal cohesion was achieved. Relatively high electrode loading (71 mg cm^{-2} , 13.2 mAh cm^{-2}) was also realized in this work.

The general impression after reading this work is that it's a good report on the evaluation of process parameters. But the concept and mechanism discussion cannot meet the quality requirement for Nat Commun.

For dry process electrode in the application of LIBs, MWNT and PVDF are common additives for conducting and binder purpose. The process used in this work, including the powder compression, hot press, and roll to roll process have all been widely reported in this field. I don't see significant improvement in this aspect.

The authors also compared the morphology difference between the dry process electrode and slurry coating ones. It's normal that the loading of active materials in dry process is much higher than slurry process, but it doesn't mean electrode in dry process can outperform the slurry process.

In the performance evaluation, the cycle performance of the slurry process electrode drops 70% in only 400 cycles. This doesn't represent the true performance of normal SCEs, and is misleading. The authors also present the pouch cell results, and give the energy density of the pouch cell. I suggest the authors remove the energy density data at 370 Wh/kg , the weight need to be derived from the whole cell mass that weighted of the accrual cell instead of estimation. Otherwise, the energy density result is also misleading.

Reviewer #2 (Remarks to the Author):

This manuscript reported fabrication of thick electrodes with dry press process and showed promising results. Dry processing poses significant benefits in reducing manufacturing cost and environmental impact. This is an interesting area to explore. However, the work was not well presented and can't be published in current version. Below are some comments:

1) Some of the references are not very suitable. For example, in page 4 line 63, more appropriate references could include Journal of Power Sources 466, 2020, 228315 and ACS Sustainable Chemistry & Engineering 8 (8), 2020, 3162-3169.

2) The motivation of using MWNT should be provided in the introduction. The sentences in page 6 line 103 is better suited for introduction.

3) In page 11 line 231-236, the better cohesion in DPCE is not only due to the MWNT but also the higher density (better contact).

4) The abbreviation should be spelled out or defined when 1st time occurs. For example, what's DPCE 1505?

5) The work was not well presented. There were so many different electrodes. It is hard to track which one was from each figure.

6) When comparing the DPCE and SCE electrodes, did they have same mass loading? Was the SCE electrode calendered?

7) The formulation for the SCE was not appropriate. The binder content was too low for a 15% conductive additive, which contributes to the high EIS.

8) What was 1C defined?

9) In Fig. 7, the electrodes seemed too thick? What's the relative density or porosity of those

electrodes? The porosity seems very high.

10) When talked about cell energy density, the pouch material, tabs, etc. should also be considered. In addition, the average voltage may not be 3.8 V anymore. The authors should use the energy from cell testing for the calculation.

RESPONSE TO REVIEWERS' COMMENTS

Reviewer #1

This work reports a dry press-coating technique to fabricate the high loading electrode for lithium-ion batteries. The authors use MWNT / PVDF composite scaffold as the active material host and an etched Al foil as a current collector. With the mixing, casting, hot press, and roll process, the thick electrode with strong internal cohesion was achieved. Relatively high electrode loading (71 mg cm^{-2} , 13.2 mAh cm^{-2}) was also realized in this work. The general impression after reading this work is that it's a good report on the evaluation of process parameters. But the concept and mechanism discussion cannot meet the quality requirement for Nat Commun.

For dry process electrode in the application of LIBs, MWNT and PVDF are common additives for conducting and binder purpose. The process used in this work, including the powder compression, hot press, and roll to roll process have all been widely reported in this field. I don't see significant improvement in this aspect.

The authors also compared the morphology difference between the dry process electrode and slurry coating ones. It's normal that the loading of active materials in dry process is much higher than slurry process, but it doesn't mean electrode in dry process can outperform the slurry process.

In the performance evaluation, the cycle performance of the slurry process electrode drops 70% in only 400 cycles. This doesn't represent the true performance of normal SCEs and is misleading. The authors also present the pouch cell results and give the energy density of the pouch cell. I suggest the authors remove the energy density data at 370 Wh/kg , the weight need to be derived from the whole cell mass that weighted of the accrual cell instead of estimation. Otherwise, the energy density result is also misleading.

→ The authors would like to first appreciate the reviewer for raising the concerns, which we believe is caused by misunderstandings of the core propositions of our manuscript. Please thus allow us to clarify. The main point of our work is to demonstrate that with the proposed dry press-coating technique, a breakthrough performance and brand-new manufacturing concept of the dry LIB electrode can be achieved, that is, a scalable ultrahigh electrode loading (71 mg cm^{-2} , 13.2 mAh cm^{-2}) lithium metal battery with a superb energy density of 763 Wh L^{-1} . The dry MWNT-

PVDF composite uniquely forms a robust and uniform coating layer on the etched Al current collector with a simple hot-pressing method, which is a previously unseen phenomena of MWNT in a solvent-free medium. The solid-solid adhesion mechanism introduced in other dry LIB electrode reports necessitate the help of adhesion inducing agents or techniques such as the holey graphene (HG), paraffin wax, electrostatic spray deposition (ESD), spray drying technique, gravure printing method, etc. However, the use of such adhesion inducing agents or techniques pose various limitations like a low mass loading, weak cohesion, delamination upon bending, negligible flexibility, scalability issue, etc. The ultrahigh loading full operational DPCE distinguishes it from all previous dry processed electrodes that show a loading threshold way below the standard we set here, for example, with no more than 70 mg cm⁻² mass loading or only 10 mAh cm⁻² areal capacity (Advanced Materials Technologies 2.10 (2017): 1700106). We have recently achieved a new record of 92 mg cm⁻² mass loading DPCE with a remarkable areal capacity of 16 mAh cm⁻². A comparison with recent refs [1-6] is shown in the table below:

Reference	1	2	3	4	5	6	This work
Electrode preparation method	Dry cold press (20-25 MPa)	Dry cold press (20-500 MPa)	Electrostatic spray deposition + Hot roll press	Electrostatic spraying + roll press	Continuous molding + hot roll press	Cold plasma process powder coating	Dry press-coating (10 MPa)
Preparation time	5 min	10 min	1 min + 60 min	30 s + 120 min	10 s + 60 min	3 min	30 s
Process temperature (°C)	RT	RT	170	200	180	Not specified	180
Active material (wt %)	NCM523 (77%)	LFP (50%)	NCM111 (90%)	NCM523 (87.5%)	NCM (90%)	LCO (~ 60%)	NCM712 (80 %)
Conductive agent	Holey graphene	Holey graphene	Carbon black	Carbon black	Carbon black	Al particle	MWNT
Binder			PMMA	PVDF	PVDF		PVDF
Current collector	Al foil	Al foil	Electrically grounded Al foil	Electrically grounded Al foil	Electrically grounded Al foil	Al foil	Etched Al foil
Mass loading (mg cm ⁻²)	26	11	10	3	65	42	92
Areal capacity (mAh cm ⁻²)	3.2 (0.1 C)	1.9 (0.1 C)	1.5 (0.2 C)	0.5 (1.0 C)	9.1 (0.1 C)	4 (0.2 C)	16 (0.1 C)

* The highlighted sections indicate the distinguishing features and performances of DPCE from other dry-processed electrodes

1. Walker, B. A. et al. Dry-pressed lithium nickel cobalt manganese oxide (NCM) cathodes enabled by holey graphene host. *Electrochim. Acta* **362**, 137129 (2020).
2. Kirsch, D. J. et al. Scalable Dry Processing of Binder-Free Lithium-Ion Battery Electrodes Enabled by Holey Graphene. *ACS Appl. Energy Mater.* **2**, 2990–2997 (2019).
3. Al-Shroofy, M. et al. Solvent-free dry powder coating process for low-cost manufacturing of LiNi_{1/3}Mn_{1/3}Co_{1/3}O₂ cathodes in lithium-ion batteries. *J. Power Sources* **352**, 187–193 (2017).
4. Zhen, E. et al. Effects of binder content on low-cost solvent-free electrodes made by dry-spraying manufacturing for lithium-ion batteries. *J. Power Sources* **515**, 230644 (2021).
5. Liu, J. et al. Scalable Dry Printing Manufacturing to Enable Long-Life and High Energy Lithium-Ion Batteries. *Adv. Mater. Technol.* **2**, 1700106 (2017).
6. Liang, Z, et al. Solvent-Free Manufacturing of Lithium-Ion Battery Electrodes via Cold Plasma. *Energy Environ. Mater.* 10.1002/eem2.12503 (2022) [in press].

Another point to emphasize is the fact that unlike other dry processed electrode papers, the demonstration of the ultrahigh loading DPCE highlighted at the end of the result and discussion section was not contrived by sacrificing the scalability or performance aspect of DPCE. Normally, there exists a trade-off relationship between the electrode mass loading and the adhesion with the current collector because of the maximum threshold of adhesive force at the interlayer as well as the cohesive force between the electrode particles. Furthermore, the increase in the mass loading give rise to a thick and bulky electrode layer with a restricted tortuosity and high cell resistance which negatively affect the electrochemical performance of the cell. Whereas the DPCE presented in our manuscript does not have the aforementioned issues but in turn shows stronger adhesion with the current collector as the loading increases and exhibits a smooth cycling performance at a pouch-cell level. Thus, we have not only secured the mechanical property but also the electrochemical property of the DPCE which can aid towards downsizing battery packs in electric vehicles.

The reviewer mentioned that the higher loading of active materials in the dry process compared to slurry process is normal, and also raised the concern that the aforementioned factor doesn't mean that electrodes in the dry process can outperform the slurry processed ones. We would like to emphasize first that our manuscript was built upon the electrochemical performance data of DPCEs, which proves its superiority over the SCEs on various parameters such as the discharge capacity under variable current densities, a long-term cycle stability, impedance, Li-ion diffusion coefficient etc. To level the playing field, the mass loading as well as electrode composition of both DPCEs and SCEs were fixed and examined with at least 3 sample cells each. It was noteworthy that the DPCEs excelled in every electrochemical analysis compared to SCEs and achieved a higher capacity retention result even at a full-cell level (84 vs. 78%). Furthermore, the dry press-coating technique allowed the fabrication of ultrahigh loading electrodes (here it is $\geq 30 \text{ mg cm}^{-2}$) and led to breakthrough performances, which was definitely not possible to achieve with the conventional slurry-coating process. Therefore, we can say that DPCEs can set a new level of competing ground which is beyond reach with the conventional solvent processed electrodes.

Additionally, the reviewer mentioned that the dropping of cycle performance of the slurry process electrode to 70% in only 400 cycles doesn't represent the true performance of normal SCEs and is misleading. We would like to note that a great deal of effort was put into choosing the reference data for the comparison with DPCE result because we also duly acknowledge the importance of data reliability. Often, it can be seen in many papers where the performance result absurdly stands out from the reference or previously reported data without providing any supplementary information about the test condition. To depict a true performance, we thus carried out the comparative analysis of DPCE with SCE under the same condition i.e., areal mass loading, electrode composition (active material/conductive agent/binder), current density, temperature etc. We understand that there are numerous factors associated with the cell degradation in a high-nickel NMC lithium metal cell such as transition metal dissolution, lithium dendrite, electrolyte decomposition, parasitic side reactions etc., which impede long-term cycling and structural stability. Therefore, the cycle performance data of reference electrode (SCE) was extracted from at least 3 replicate sample cells, where the best result was taken and used to compare with the

performance of DPCE. Particularly, this extracted cycle performance of SCE exceeds the reference data reported in other papers as shown below:

Reference	1	2	3	4	5	6	This work
Active material	NCM523	NCM811	NCM111	NCM811	NCM111	NCM712	NCM712
Reference electrode composition (AM/CB/Binder)	87.5/7.5/5	97/2/1	90/5/5	94/3/3	80/10/10	80/10/10	80/15/5
Electrolyte	1M LiPF ₆ in 1:1 EC/DMC	1M LiPF ₆ in 3:7 EC/EMC + 10 wt% FEC and 2 wt% VC	1M LiPF ₆ in 1:1:1 EC/DEC/DMC	1M LiPF ₆ in 3:7 EC/EMC + 2 wt% VC	1M LiPF ₆ in 3:7 EC/EMC	1M LiPF ₆ in 1:1 EC/DMC	1M LiPF ₆ in 1:1 EC/DEC + 10 wt% FEC
Fabrication method	Slurry coating process						
Mass loading	Not specified	~3.5 g cc ⁻¹	Not specified	~7.5 mg cm ⁻²	Not specified	10 mg cm ⁻²	~9 mg cm ⁻²
Current density (C-rate)	1 C	0.5 C	0.5 C	0.5 C	1 C	1 C	1 C
Voltage range	3.0-4.4 V	3.0-4.2 V	2.8-4.3 V	3.0-4.3V	3.0-4.6 V	2.8-4.3 V	2.9-4.3 V
Cycle number	300	240	50	300	200	100	400
Capacity retention	69%	0%	84%	37.1%	47.7%	47%	35%

1. Zhen, E. et al. Effects of binder content on low-cost solvent-free electrodes made by dry-spraying manufacturing for lithium-ion batteries. *J. Power Sources* **515**, 230644 (2021).
2. Kim, N. et al. Amphiphilic Bottlebrush Polymeric Binders for High-Mass-Loading Cathodes in Lithium-Ion Batteries. *Adv. Energy Mater.* **12**, 2102109 (2021).
3. Ludwig, B., Zheng, Z., Shou, W., Wang, Y. & Pan, H. Solvent-Free Manufacturing of Electrodes for Lithium-ion Batteries. *Sci. Rep.* **6**, 23150 (2016).
4. Xue, W., Huang, M., Li, Y., Zhu, Y., Gao, R., & Xiao, X. et al. Ultra-high-voltage Ni-rich layered cathodes in practical Li metal batteries enabled by a sulfonamide-based electrolyte. *Nat. Energy* **6**, 495-505 (2021).
5. Xu, G., Liu, Q., Lau, K., Liu, Y., Liu, X., & Gao, H. et al. Building ultraconformal protective layers on both secondary and primary particles of layered lithium transition metal oxide cathodes. *Nat. Energy* **4**, 484-494 (2019).
6. Wang, M., Wang, J., Si, J., Chen, F., Cao, K., & Chen, C. Bifunctional composite separator with redistributor and anion absorber for dendrites-free and fast-charging lithium metal batteries. *Chem. Eng. J.* **430**, 132971 (2022).

The comparison with other reference electrodes clearly shows that the cycle performance of SCE is not an underrated result but rather a challenging data for the juxtaposition, as it outperforms all other reference electrodes even at a much higher mass loading. Although, an exact one-to-one comparison against SCE condition could not be done due to a lack of prior example, but we tried to find as many reference electrode data as possible from high impact journals, even including those that were tested under similar but milder conditions to verify our result. As mentioned earlier the performance data of SCE was extracted from the best outcome of a group of replicate sample cells, and it was done to accurately depict the DPCE's performance and more so not to overemphasize its true electrochemical properties. Therefore, we firmly believe that our SCE's cycle performance is a reasonable parameter for the comparative analysis.

In regard to the comment that the energy density of the pouch cell needs to be recalculated based on the whole cell mass, the authors genuinely agree with the reviewer and also feel the needs to make amendments as advised. However, the reason why we excluded the mass of the packaging

Editorial Note: First image below from Park, SH., King, P.J., Tian, R. et al. High areal capacity battery electrodes enabled by segregated nanotube networks. *Nat Energy* **4**, 560–567 (2019), reproduced with permission from Springer Nature. Second image below from Wu, X., Xia, S., Huang, Y., Hu, X., Yuan, B., Chen, S., Yu, Y., Liu, W., High-Performance, Low-Cost, and Dense-Structure Electrodes with High Mass Loading for Lithium-Ion Batteries. *Adv. Funct. Mater.* 2019, 29, 1903961, reproduced with permission from John Wiley and Sons.

materials of the pouch cell in the first place was because many previous works on high loading electrodes calculated the energy densities without considering the cell package. Below are some of the captured images from the previously reported works:

Supplementary Note 1.

As shown in Fig. 5e of the main text, all data is in close agreement with the dashed lines plotted from the equation relating E_{SP} to C/A . This equation could be derived according to,

$$E_{SP} = \frac{E}{M_{Total_cell}} = \frac{E/A}{M_{Total_cell}/A} = \frac{E/A}{\frac{M_{Cathode}}{A} + \frac{M_{Anode}}{A} + \frac{M_{Inactive}}{A}} \quad [Eq.1]$$

where $M_{Cathode}$, M_{Anode} , and $M_{Inactive}$ are mass for cathode, anode and inactive components (Al/Cu foils, separator and electrolyte filled in cathode/anode and separator pores), respectively.

Here, E , the cell energy, can be described as follows,

$$E = \int V(t) \cdot Idt \approx V \cdot C_{Cell}$$

[Eq.2]

where V and C_{Cell} are the average operating voltage of the cell and the cell capacity, respectively.

Figure. A captured image from a research article published in *Nat. Energy*, **4**, 560, (2019), “High areal capacity battery electrodes enabled by segregated nanotube networks”. The energy densities (here, E_{sp}) were calculated excluding the packaging materials.

Supplementary Note 1. Calculation specific energy density of the full cell.

Specific energy density ($Wh\ kg^{-1}$):
 $10^3 \times \text{Areal capacity (mAh cm}^{-2}\text{)} \times \text{Electrode area (cm}^2\text{)} \div \text{Total weight of electrode (mg, anode + cathode)} \times \text{Average operating voltage (V)}$

Figure. A captured image from a research article published in *Adv. Funct. Mater.*, **29**, 1903961, (2019), “High-Performance, low-cost, and dense-structure electrodes with high mass loading for

Editorial Note: Table below reproduced from Xue, W., Huang, M., Li, Y. et al. Ultra-high-voltage Ni-rich layered cathodes in practical Li metal batteries enabled by a sulfonamide-based electrolyte. *Nat Energy* **6**, 495–505 (2021), reproduced with permission from Springer Nature.

lithium-ion batteries”. The specific energy density was calculated without including the coin cell packaging materials.

Supplementary Table 4 Parameters used for calculating the specific energy of the single-layer pouch cell

	Weight (mg)
Cathode	109.9
Anode	47.9
Separator	6.1
Electrolyte	43.5
Total weight (mg)	207.4

Figure. A captured image from a research article published in *Nat. Energy*, **6**, 495-505, (2021), “Ultra-high-voltage Ni-rich layered cathodes in practical Li metal batteries enabled by a sulfonamide-based electrolyte”. The specific energy density was calculated without including the pouch cell packaging materials.

Note that the energy densities were calculated based on just the inner materials, without including the packaging components. Therefore, to make a fair comparison with those of previously reported high-loading electrodes, the energy densities of the DPCE pouch cell were calculated without including the packaging materials.

However, the more accurate expression of energy density is unarguably the one that includes the entire mass of cell. Also, we believe that such representation can give our result an edge over others. Therefore, we recalculated the specific energy and volumetric energy density values based on the weight and volume of the accrual cell (including the cell package) and provided the information in **Fig. 6e,f** and **Supplementary Table 1** of the revised manuscript as shown below. Note that additional pouch cell performance results with various other areal mass loading DPCEs have also been included in the revised manuscript (the calculation result in **Supplementary Table 1** represents the energy density of the Li-metal pouch cell assembled with the highest areal mass loading DPCE (100 mg cm⁻²)).

[Revised manuscript]

Fig. 6 | The ultrahigh mass loading DPCE (Pouch-cell). **a**, Charge/discharge voltage profile of the Li-metal DPCE pouch cell (31 mg cm^{-2}) at a voltage range of 2.9-4.3 V and charge/discharge current density of 0.1/0.1 C (dashed line represents the theoretical capacity of NCM712). **b**, Cycling performance of the Li-metal DPCE pouch cell (31 mg cm^{-2}) at a charge/discharge current

density of 0.5/0.5 C at a voltage range of 2.9-4.3 V. **c**, Charge/discharge voltage profiles of the Li-metal DPCE pouch cells in terms of areal mass loadings at a voltage range of 2.9-4.3 V and charge/discharge current density of 0.05/0.1 C. **d**, Cycling performance of the Li-metal DPCE pouch cells with various areal mass loadings. **e**, Areal capacities of cells as a function of areal mass loading (DPCEs versus previously reported solvent-free cathodes). **f**, Specific energies (calculated based on the entire cell weight) of cells as a function of areal capacity (DPCEs versus previously reported high mass loading cathodes).

“Given this ultrahigh loading design, a high energy density Li-metal pouch cell can be fabricated, and the calculated specific energy and volumetric energy density (based on the entire cell weight) were up to 360 Wh kg⁻¹ and 701 Wh L⁻¹, respectively at SOC 0% (Supplementary Table 1). It is noteworthy that this ultrahigh loading capability of the DPCE pouch cell far exceeded that of the previously reported solvent-free electrodes in terms of both areal capacity and mass loading (Fig. 6e, Supplementary Table 2), and even surpassing other high loading cathodes fabricated with different methods (Fig. 6f, Supplementary Table 3).”

Supplementary Table 1 | Calculation details for the specific energies and volumetric energy densities of the Li metal cells equipped with the DPCEs.

The specific energy of the Li metal cell was calculated based on the following equation:

$$\begin{aligned} \text{Specific energy (Wh kg}^{-1}\text{)} &= \frac{\text{Energy}}{\text{Mass of cell}} = \frac{\frac{\text{Energy}}{\text{Area}}}{\frac{\text{Mass of cell}}{\text{Area}}} \\ &= \frac{\text{Nominal voltage} \times C/A}{M_{\text{cathode}}/A + M_{\text{anode}}/A + M_{\text{separator}}/A + M_{\text{electrolyte}}/A + M_{\text{package}}/A} \end{aligned}$$

where M_{cathode} , M_{anode} , $M_{\text{separator}}$, $M_{\text{electrolyte}}$ and M_{package} denote the mass of cathode (including the etched Al current collector (20 μm)), anode (comprised of Li metal (200 μm) and Cu current collector (18 μm)), PE separator, injected electrolyte, and **pouch-cell package** respectively (C and A refer to the capacity and area, respectively).

C/A [mAh cm ⁻²]	Nominal voltage [V]	M_{cathode}/A [mg cm ⁻²]	M_{anode}/A [mg cm ⁻²]	$M_{\text{separator}}/A$ [mg cm ⁻²]	$M_{\text{electrolyte}}/A$ [mg cm ⁻²]	M_{package}/A [mg cm ⁻²]	M_{total}/A [mg cm ⁻²]	Specific energy [Wh kg ⁻¹]
17.6	3.9	105.4	26.8	0.8	36.0	21.3	190.3	360

The volumetric energy density of the Li metal cell was calculated based on the following equation:

$$\text{Volumetric energy density (Wh L}^{-1}\text{)} = \frac{\text{Energy}}{\text{Thickness of cell}} = \frac{\text{Nominal voltage} \times \text{C/A}}{T_{\text{cathode}} + T_{\text{anode}} + T_{\text{separator}} + T_{\text{package}}}$$

where T_{cathode} , T_{anode} , $T_{\text{separator}}$ and T_{package} are the thickness of cathode, anode, PE separator and **pouch-cell package**, respectively.

C/A [mAh cm ⁻²]	Nominal voltage [V]	T_{cathode} [μm]	T_{anode} [μm]	$T_{\text{separator}}$ [μm]	T_{package} [μm]	T_{total} [μm]	Volumetric energy density [Wh L ⁻¹]
17.6	3.9	593	218	18	150	979	701

Reviewer #2

This manuscript reported fabrication of thick electrodes with dry press process and showed promising results. Dry processing poses significant benefits in reducing manufacturing cost and environmental impact. This is an interesting area to explore. However, the work was not well presented and can't be published in current version. Below are some comments:

1) Some of the references are not very suitable. For example, in page 4 line 63, more appropriate references could include Journal of Power Sources 466, 2020, 228315 and ACS Sustainable Chemistry & Engineering 8 (8), 2020, 3162-3169.

→ Thank you so much for the reviewer's considerate comment. We agree that the advised references are far more appropriate and can add validity to our analysis. In response to the reviewer's comment, the advised references were added as ref. [17,18]. Below is the revised manuscript (Please note that some amendments have been made to the content in introduction section for better clarity and conciseness of the presentation):

[Revised manuscript]

“As a breakthrough approach, the dry process is considered a new electrode fabrication method for post-LIB electrodes because they offer unparalleled advantages in terms of operating cost and energy efficiency compared to the conventional solvent process. Moreover, the dry process can pave a path to battery miniaturization as the absence of solvent elevates the maximum threshold of active mass loading and thus allowing the fabrication of higher mass-loading electrodes¹⁴⁻¹⁸.”

References

17. Hawley, W. B., Parejiya, A., Bai, Y., Meyer, H. M., III, Wood, D. L., III, & Li, J. Lithium and transition metal dissolution due to aqueous processing in lithium-ion battery cathode active materials. *J. Power Sources* **466**, 228315 (2020).

18. Sahore, R., Wood, D. L., III, Kukay, A., Grady, K. M., Li, J., & Belharouak, I. Towards understanding of cracking during drying of thick aqueous-processed LiNi_{0.8}Mn_{0.1}Co_{0.1}O₂ cathodes. *ACS Sustainable Chem. Eng.* **8**, 3162–3169 (2020).

2) The motivation of using MWNT should be provided in the introduction. The sentences in page 6 line 103 is better suited for introduction.

→ Our deep appreciation is devoted to the reviewer's insightful comment. We also agree that the given guidance can greatly improve the logical flow of our presentation. In response to the reviewer's comment, some content restructurings were made and the motivation of using MWNT was transferred to the introduction. Below is the revised manuscript:

[Revised manuscript]

“These previous reports on the dry LIB electrode process have mainly focused on either changing the coating process or the binder to increase the performance of the dry electrode but rarely explored alternative ways, such as employing a new conductive agent or current collector, to answer the core challenges of solvent-free electrode fabrication, which include weak cohesive strength, low deformability, high cell polarization, low rate capability, etc.

Carbon nanotubes (CNTs) are among the most avidly studied and utilized materials for multipurpose LIB electrode fabrication owing to their remarkable electronic conductivity, mechanical strength, resistance to chemical degradation, etc^{22,23}. To the best of our knowledge, only little information can be found on the use of dry powdered CNTs along with a polymeric binder to directly press-coat electrode material onto a current collector in a completely solvent-free approach. Therefore, the dry press-coating capability of the MWNT-PVDF composite powder was evaluated for the first time by measuring its adhesive and cohesive strength upon pressing. In addition, etched Al foil was selected as a new current collector (Supplementary Fig. 1) to enhance adhesion by inducing an anchoring effect on the submicron pores of the foil surface. It was reported that a larger contact area with the Al₂O₃ passive layer improves the connection of the electrode

film with the substrate surface²⁴⁻²⁶. As shown in the X-ray photoelectron spectroscopy (XPS) results (Supplementary Fig. 2), the Al 2p photoelectron spectra indicate a higher amount of the Al₂O₃ layer on the etched Al foil than on the normal Al foil.

Herein, we developed a novel method to fabricate a solvent-free LiNi_{0.7}Co_{0.1}Mn_{0.2}O₂ (NCM712) electrode, namely, a dry press-coated electrode (DPCE), via a facile one-step hot-pressing of premixed NCM712, multiwalled carbon nanotubes (MWNTs), and a dry PVDF powder mixture onto etched Al foil (Fig. 1 and Supplementary Fig. 3). Additionally, the influences of the MWNT and binder content on the electrode structure and electrochemical performance were also studied. The DPCE with the optimal composition was then compared with conventional slurry-coated electrodes (SCE) on various aspects, such as morphology and electrochemical performance. Furthermore, to manifest an excellent 3D conductive network, ultrahigh-loading DPCE pouch cells were also fabricated.”

3) In page 11 line 231-236, the better cohesion in DPCE is not only due to the MWNT but also the higher density (better contact).

→ Many thanks for the reviewer’s constructive comment. As the reviewer pointed out the better cohesion in DPCE is not only due to the MWNT but also the higher density (better contact) and more compact electrode structure. However, the analysis of DPCE’s cohesion property is not dealt with in the guided section (page 11 line 231-236) but presented in the result section (page 7 line 136-146) of the original manuscript (as shown below). It is assumed that more explanation about the cohesion property should be included, and we also think the original explanation is not sufficient. Hence, in response to the reviewer’s comment, a more in-depth evaluation of DPCE’s cohesion property was made and provided in the revised manuscript as below:

[Original manuscript]

<Page 11 line 231-236>

“(Supplementary Fig. 10). When the electrodes were immersed in the electrolyte solution and ultrasonicated for 3 min, the DPCE maintained its original structure, and the electrolyte solution remained clear. In contrast, the SCE incurred delamination due to exfoliation on the surface, and the electrolyte solution became turbid. This result indicates the presence of a robust 3D conductive framework in the DPCE along with abundant nanochannels for rapid ion transport.”

<Page 7 line 136-146>

“Additionally, an in-house pull-off test was carried out to emphasize the synergistic effects between MWNT and PVDF, as illustrated in Fig. 2e. The pull-off test is a commonly practiced method that measures the adhesive strength of a substrate coating; here, it was used to analyze the cohesive properties of the scaffold by measuring the tensile strength after stripping the attached etched Al foils in the opposite z-axis directions.^{28,29} As seen in Fig. 2f, the tensile strength of the sandwiched d-MP electrode was much greater than that of d-SP and even surpassed that of the sandwiched electrode containing bare PVDF dry powder. The notable tensile strength of the d-MP electrode was attributed to the intertwined MWNT-PVDF network that spans the entire electrode body.³⁰ Therefore, the MWNT-PVDF composite exhibits both favorable adhesive and cohesive properties, which makes it an excellent ingredient for the dry electrode fabrication process.”

[Revised manuscript]

“Additionally, to investigate the cohesive property of MWNT-PVDF composite, the pull-off test, which is commonly used for measuring the bond strength of a substrate coating was conducted^{28,29}; but here, it was modified to examine the cohesive strength by measuring the force required to completely pull apart the sandwiched dry electrode, made by hot-pressing different dry powder samples between two etched Al foils. By this means, three different sandwiched dry electrodes were prepared using MWNT-PVDF (d-MP), PVDF, and Super P-PVDF (d-SP) powder, and their respective tensile strengths were comparatively measured by stripping the electrodes in the opposite z-axis directions (Fig. 2e). Remarkably, the tensile strength of d-MP sandwiched dry

electrode far exceeded that of both d-SP and PVDF sandwiched dry electrodes, indicating a presence of superior cohesion and synergistic interaction between MWNT and PVDF (Fig. 2f). This outstanding tensile strength of the d-MP electrode can be attributed to the tightly intertwined MWNT-PVDF network which is capable of achieving the nano hook-and-loop fastening effect within the electrode body and concatenate the entire structure³⁰. Evidently, the MWNT-PVDF composite exhibits both favorable adhesive and cohesive properties in conjunction with the etched Al foil, and thus it was chosen as a main ingredient for the DPCE fabrication in this work.”

4) The abbreviation should be spelled out or defined when 1st time occurs. For example, what’s DPCE 1505?

→ We thank the reviewer for the thoughtful and considerate comment. As the reviewer mentioned, the abbreviation should be spelled out or defined when it appears for the first time, and we did so whenever the new abbreviations were introduced in the main body. However, for conciseness and brevity of the manuscript, explanation about the abbreviated names of DPCE samples i.e., DPCE 1505, DPCE 1010 and DPCE 0515 was presented in the methods section rather than in the main body. We thought that the abbreviations of different DPCE samples can be more elaborately and coherently explained along with the DPCE preparation procedure in the methods section. Thus, we reinforced the methods section and provided in the revised manuscript below:

[Revised manuscript]

“Effects of the MWNT and PVDF binder contents on the mechanical and electrochemical properties of the DPCE.

Given the applicability of MWNT-PVDF composite in DPCE fabrication, the effects of MWNT and PVDF content on the mechanical and electrochemical properties of the DPCE were investigated. First, the mechanical properties of **DPCE samples with varying MWNT and PVDF ratio (see Methods section for details)** were evaluated using a 180° peel test. According to the peel test results (Supplementary Fig. 5a), both DPCE 0515 (4.1 N cm⁻¹) and DPCE 1010 (4.0 N cm⁻¹)

showed higher average adhesion forces than DPCE 1505 (3.2 N cm^{-1}), which adheres to the logic that an increase in binder content enhances the adhesive strength with the current collector. As displayed in the stripped tape images (Supplementary Fig. 5b), detachment of the electrodes becomes less severe as the binder content increases. However, there was no significant decrease in the adhesive strength of the electrode as the binder content decreased from 15 to 10 wt.%, demonstrating that the corresponding increase in MWNT content (5 to 10 wt.%) creates a more concrete and supportive scaffold that helps the electrode layer cling better onto the surface of the etched Al foil.”

“**Methods**

DPCE fabrication. NCM712, MWNTs, and PVDF binder were premixed and ground using a mortar. The amount of active material was fixed at 80 wt.%, while the amount of MWNTs and PVDF binder were varied with the overall weight ratio compositions of 80/15/5, 80/10/10, and 80/5/15. **The as-prepared DPCEs were named DPCE 1505, DPCE 1010, and DPCE 0515 (number refers to the weight ratio of MWNT and PVDF).** All powders were dried in a vacuum oven at $80 \text{ }^\circ\text{C}$ before use.”

5) The work was not well presented. There were so many different electrodes. It is hard to track which one was from each figure.

→ We thank the reviewer for raising this concern and for giving necessary guidelines as to how we can improve the conciseness and clarity of our manuscript. Therefore, portion of the main body especially the part that deals with the DPCE optimization was moved to the supplementary information. In this way, the manuscript can be more condensed and focus more on the comparative analysis with the SCEs. In response to the reviewer’s comment, a comprehensive structural modification of the manuscript was made and provided in the revised manuscript below:

[Revised manuscript]

“Finally, the optimization of the DPCE composition was carried out by comparatively evaluating the rate capability, cycle performance, electrochemical impedance spectra (EIS), and charge-discharge profiles of the three different DPCEs using a half-cell test (**Supplementary Note 1 and Supplementary Fig. 9**). It was found that DPCE 1505 composition (NCM712/MWNT/PVDF – 80/15/5) exhibited the best performance among all, which confirms the significance of MWNT and PVDF ratio on the electrochemical performance of DPCEs. Furthermore, using the optimal composition, DPCEs with various other active materials, such as NCM622, LCO, and LFP, were also fabricated, all of which exhibited excellent rate capabilities (Supplementary Fig. 10), verifying the versatile nature of the dry press-coating process.”

“**Supplementary Note 1 | Optimization of the DPCE composition**”

In order to optimize DPCE composition, the DPCEs with varying MWNT and PVDF binder contents, namely DPCE 1505 (NCM712/MWNT/PVDF – 80/15/5), DPCE 1010 (NCM712/MWNT/PVDF – 80/10/10) and DPCE 0515 (NCM712/MWNT/PVDF – 80/5/15) were prepared and investigated using a half-cell test (Supplementary Fig. 9). As shown in Supplementary Fig. 9a, DPCE 1505 displayed better discharge rate capability over DPCE 1010 and DPCE 0515. Particularly, DPCE 0515 which has the highest PVDF content showed a noticeable deterioration in rate capability starting from 1.0 C, and further becoming worse as the rate increased from 2.0 C to 5.0 C. Moreover, in the long-term cycling performance result (Supplementary Fig. 9b), DPCE 1505 showed the highest initial reversible discharge capacity of 176.7 mAh g⁻¹ and a capacity retention of 77.7% after 250 cycles at 0.5 C, which exceeded the performances of the other two electrodes. To further analyze the electron/ionic transportation behaviors of the electrodes, electrochemical impedance spectroscopy (EIS) experiments were performed. As shown in the Nyquist plots (Supplementary Fig. 9c), which include the real-axis intercept corresponding to the ohmic resistance (R_o) and a single depressed semicircle in the high-medium frequency region corresponding to the charge transfer resistance (R_{ct}), the DPCE 1505 showed much smaller R_o (2.2 Ω) and R_{ct} (54 Ω) values compared to DPCE 1010 (2.4 and 64 Ω) and DPCE 0515 (4.2 and 88 Ω). Additionally, the charge-discharge profiles of the three DPCEs

during long-term cycling (Supplementary Fig. 9d-f) manifest the significantly small capacity decay and overpotential increase in DPCE 1505 over other electrodes. Therefore, based on these results, the DPCE 1505 (NCM712/MWNT/PVDF – 80/15/5) was chosen as a representative cathode.

Supplementary Figure 9 | The electrochemical performance of the DPCE with varying MWNT and PVDF content. **a**, Discharge rate capability over a range of different current densities (0.2-5.0 C). **b**, Cycle performance and corresponding coulombic efficiency at 0.5 C. **c**, Nyquist plots of pristine DPCEs. **d-f**, Charge/discharge voltage profile of DPCE 1505 (**d**), DPCE 1010 (**e**) and DPCE 0515 (**f**).

6) When comparing the DPCE and SCE electrodes, did they have same mass loading? Was the SCE electrode calendered?

→ Many thanks for the reviewer's valuable comment. In order to make a fair comparison between DPCE and SCE, the areal mass loadings of electrodes were made the same, and both the electrodes

were calendered before use. It is known that the mass loading of electrode is an important factor that affects the mechanical property and electrochemical performance, fixing the areal mass loading is crucial when comparing the intrinsic properties between different electrode samples. Consequently, in the original manuscript, information about the areal mass loadings was mentioned when comparing between DPCE and SCE as shown below. However, for the case of high loading SCEs, it was not possible to fabricate beyond certain mass loading nor calender via roll press due to the structural instability and delamination caused by the solvent evaporation during the drying process. Hence, the highest attainable mass loading SCE was compared with the high loading DPCE.

[Original manuscript]

“The electrochemical properties of the DPCE and SCE are displayed in Fig. 3. DPCE clearly displays a better rate capability than SCE at all current densities (Fig. 3a). A long-term cycling test was also performed using a half-cell at 1.0 C **with a mass loading of both electrodes at 8-9 mg cm⁻²** (Fig. 3b). The DPCE demonstrated much better cycling stability with an initial capacity of 170 mAh g⁻¹ (1.0 C) and capacity retention of 67% after 400 cycles along with stable coulombic efficiency. In contrast, the SCE delivered an initial capacity of only 159 mAh g⁻¹ (1.0 C) with the much lower capacity retention of 35% after 400 cycles and unstable coulombic efficiency after 300 cycles. This result can be explained by post-mortem analysis of the coin cell after cycling, wherein DPCE retains its original structure with negligible cracks as opposed to SCE, which showed detachments with noticeable voids around the active materials after cycling (Supplementary Fig. 14).”

7) The formulation for the SCE was not appropriate. The binder content was too low for a 15% conductive additive, which contributes to the high EIS.

→ Many thanks for the reviewer for mentioning these very reasonable concerns. The formulation of the SCE (NCM712/Super P/PVDF – 80/15/5) was selected as such, primarily to compare with

Editorial Note: Image below reprinted from Zhen, E. et al. Effects of binder content on low-cost solvent-free electrodes made by dry-spraying manufacturing for lithium-ion batteries. *J. Power Sources* **515**, 230644 (2021), with permission from Elsevier.

the optimized DPCE (i.e., DPCE 1505), which is composed of NCM712/MWNT/PVDF – 80/15/5. The same electrode composition was used for DPCEs and SCEs because the ratio of electrode materials (active material (AM), conductive agent (CA), and binder (B)) has a significant impact on the mechanical and electrochemical properties of electrode. Moreover, to conduct a comparative analysis based on the intrinsic property of electrode, it is important to rule out the effect of the electrode composition which can potentially lead to a variation in overall properties. For this reason, most papers use the reference electrode that have the same composition with their working electrode for the comparative analysis. Some of these examples are shown below:

Figure. A captured image from a research article published in *J. Power Sources*, **515**, 230644, (2021), “Effects of binder content on low-cost solvent-free electrodes made by dry-spraying manufacturing for lithium-ion batteries”. The same electrode composition (AM:CA:B - 87.5:7.5:5)

Editorial Note: Image below reprinted from Wang, X. et al. Graphene-decorated carbon-coated LiFePO₄ nanospheres as a high-performance cathode material for lithium-ion batteries. *Carbon*, **127**, 149-157, (2018), with permission from Elsevier.

of the optimized working electrodes (solvent-free electrodes) were used to fabricate the reference electrode (conventional wet electrodes).

2.2.2. Electrochemical measurements

The working electrodes were fabricated by using LFP@C or LFP@C/G as the active material, conductive carbon black (Super-P) and polyvinylidene fluoride (PVDF) binder in a weight ratio of 80:15:5. They were mixed in a mortar for 1 h and then dispersed in N-methyl-2-pyrrolidone (NMP) solvent to form a homogeneous slurry. The slurry was coated on aluminum foil and dried in a vacuum oven at 120 °C for 10 h. The electrochemical measurements were carried out using CR2032 coin type cells, which were assembled into half-battery in an argon-filled glove box under concentrations of moisture and oxygen below 1 ppm. Lithium pellets were used as the counter/reference electrodes and Celgard 2400 was used as separator. The electrolyte solution was 1 M LiPF₆ dissolved in a mixture of ethylene carbonate (EC), dimethyl carbonate (DMC) with a volume ratio of EC: DMC = 1:1. The loading of the active mass was approximately 1.5–2.0 mg in LFP@C and LFP@C/G electrodes without deducting the mass of carbon and graphene. Fig. S1† shows illustration of the preparation process of working electrodes of LFP@C and LFP@C/G. Cyclic voltammetry (CV)

Figure. A captured image from a research article published in *Carbon*, **127**, 149-157, (2018), “Graphene-decorated carbon-coated LiFePO₄ nanospheres as a high-performance cathode material for lithium-ion batteries”. Here, two working electrodes (LFP@C and LFP@C/G) were compared against each other based on the same electrode composition (AM:CA:B – 80:15:5).

Editorial Note: Image below reprinted from Senthil, C. et al. Thermochemical conversion of eggshell as biological waste and its application as a functional material for lithium-ion batteries. *Chem. Eng. J.*, **372**, 765-773, (2019), with permission from Elsevier.

3. Electrode preparation and electrochemical testing

The electrode comprises of Ca-NCM811 as an active material, denka black as conductive carbon, and polyvinylidene difluoride (PVDF) as a binder in the respective weight ratio 80:15:5. The electrode constituents were thoroughly mixed and ground before adding 1 mL of N-methylpyrrolidone (NMP) as a solvent and homogenized to obtain a viscous slurry which was then coated over a battery grade aluminum foil (10 μm). The coated aluminum foil was dried naturally and subsequently vacuum dried at 120 $^{\circ}\text{C}$ for about 6 h to be devoid of moistures. The dried foil was cut into round disks to serve as a working electrode in the half-cell configurations. The electrochemical testing was carried out by constructing coins of CR2032-type half cells inside an argon filled glove box (Korea Kiyon, 021AS) maintained at moisture and oxygen level < 5 ppm. The coin cell assembly comprised a CaO-NCM811 disk as the working electrode, pure-cut lithium foil as the reference electrode, and Celgard 2340 microporous membrane as a separator between the electrodes. Finally, the electrode stack was saturated with an electrolytic solution comprising of one mole of lithium hexafluorophosphate (LiPF_6) salt dissolved in ethylene carbonate (EC) and dimethyl carbonate (DMC) at 1:1 v/v ratio. Electrochemical measurements, such as galvanostatic charge-discharge and rate studies, were recorded using multichannel battery cycler (Arbin), electrochemical impedance spectroscopy studies were conducted using a multichannel electrochemical analyzer (COMPACTSTAT, IVIUM technologies). All the electrochemical characterizations were performed at room temperature ($\sim 25^{\circ}\text{C}$), unless otherwise stated.

Figure. A captured image from a research article published in *Chem. Eng. J.*, **372**, 765-773, (2019), “Thermochemical conversion of eggshell as biological waste and its application as a functional material for lithium-ion batteries”. Here, the working electrode (CaO-NCM811) and reference electrode (Ca-NCM811) have the same electrode composition (AM:CA:B – 80:15:5).

Regards to the concern about the binder content being too low for a 15% conductive agent and its contribution to the high EIS result, we would like to point out that many reports including those mentioned above are using the same electrode composition as the SCE (AM:CA:B – 80:15:5) for their reference electrodes. Although, the binder content is relatively lower than the conductive agent, there was no issue during the electrode fabrication process, and more importantly, when we compared the SCE (here it is named SCE 1505) against the electrode with a higher binder content (AM:CA:B – 80:10:10, here it is named SCE 1010) using EIS, the SCE 1505 exhibited a lower charge transfer resistance compared to SCE 1010 (196 vs. 205 Ω) as shown in the figure below.

This result is due to the presence of more electrically insulating materials (polymer binder) in SCE 1010, and thus the higher binder content can negatively affect the electrochemical performance of electrodes. Consequently, the SCE composition of our manuscript (AM:CA:B – 80:15:5) can give rise to a better reference electrode and hence sets a more competitive goal to achieve.

8) What was 1C defined?

→ Many thanks for the reviewer’s valuable comment. The 1 C-rate was defined as the current density necessary to fully charge/discharge the LIB in 1h. Please note that the main active material used in our work i.e., NCM712 had a specific capacity of 160 mAh g⁻¹ at 1 C. We apologize that some of the specific capacities of the tested electrodes at 1 C were omitted in the original manuscript. Hence, the 1 C values were included and provided in the revised manuscript as below:

[Revised manuscript]

“The electrochemical properties of the DPCE and SCE are displayed in Fig. 3. DPCE clearly displays a better rate capability than SCE at all current densities (Fig. 3a). A long-term cycling test was also performed using a half-cell at 1.0 C with a mass loading of both electrodes

at 8-9 mg cm⁻² (Fig. 3b). The DPCE demonstrated much better cycling stability with an **initial capacity of 170 mAh g⁻¹ (1.0 C)** and capacity retention of 67% after 400 cycles along with stable coulombic efficiency. In contrast, the SCE delivered an **initial capacity of only 159 mAh g⁻¹ (1.0 C)** with the much lower capacity retention of 35% after 400 cycles and unstable coulombic efficiency after 300 cycles. This result can be explained by post-mortem analysis of the coin cell after cycling, wherein DPCE retains its original structure with negligible cracks as opposed to SCE, which showed detachments with noticeable voids around the active materials after cycling (Supplementary Fig. 14).”

9) In Fig. 7, the electrodes seemed too thick? What’s the relative density or porosity of those electrodes? The porosity seems very high.

→ We thank the reviewer for mentioning these very reasonable concerns. As the reviewer mentioned, the thickness of the high loading electrodes in Fig. 7 (of the original manuscript) are definitely higher than the conventional slurry coated electrodes and therefore, the porosity should be considered as a crucial factor in the electrode design. As such, the porosity (ϵ) of the high loading electrodes was calculated based on the following equation^{1,2}:

$$\epsilon = 1 - \frac{M}{T} \left[\left(\frac{\phi_{AM}}{D_{AM}} \right) + \left(\frac{\phi_{CA}}{D_{CA}} \right) + \left(\frac{\phi_B}{D_B} \right) \right]$$

where ϵ is the porosity of the electrodes, M is the areal mass loading of the electrodes, T is the thickness of the electrode, Φ_{AM} , Φ_{CA} , and Φ_B are the mass fractions of active material (NCM712), conductive agent (MWNT), and binder (PVDF) of the electrode, respectively, and D_{AM} , D_{CA} , and D_B are the true densities of active material (NCM712), conductive agent (MWNT), and binder (PVDF) of the electrode, respectively. The true densities of NCM712, MWNT, and PVDF are 3.7, 1.7, and 1.7 g cm⁻³, respectively. The calculated porosity values are shown in the table below:

Electrode composition	NCM712/MWNT/PVDF – 80/15/5
----------------------------

Areal mass loading (mg cm ⁻²)	22	25	28	30
Thickness (μm)	98	123	176	221
Porosity (%)	25	32	46	54

The calculation result shows that porosity of the high loading electrodes varies from 25 to 54%, which indicates that the porosity increases with the increase in the areal mass loading of electrodes. However, these values are well within the average NCM cathode porosity values (30-50%)^{3,4} despite greater areal mass loading and thickness of the high loading electrodes, owing to the uniaxial pressing in the dry press-coating process that produces more compact and denser electrode structure.

1. Moroni, R., Börner, M., Zielke, L., Schroeder, M., Nowak, S., Winter, M., Manke, I., Zengerle, R., & Thiele, S. Multi-Scale correlative tomography of a li-ion battery composite cathode. *Sci. Rep.* **6**, 30109 (2016).
2. Ebner, M., Geldmacher, F., Marone, F., Stampanoni, M., & Wood, V. X-Ray tomography of porous, transition metal oxide based lithium ion battery electrodes. *Adv. Energy Mater.* **3**, 845–850 (2013).
3. Hamed, H., Yari, S., D’Haen, J., Renner, F. U., Reddy, N., Hardy, A., & Safari, M. Demystifying charge transport limitations in the porous electrodes of lithium-ion batteries. *Adv. Energy Mater.* **10**, 2002492 (2020).
4. Cunha, R. P., Lombardo, T., Primo, E. N., & Franco, A. A. Artificial intelligence investigation of NMC cathode manufacturing parameters interdependencies. *Batteries Supercaps* **3**, 60–67 (2019).

10) When talked about cell energy density, the pouch material, tabs, etc. should also be considered. In addition, the average voltage may not be 3.8 V anymore. The authors should use the energy from cell testing for the calculation.

→ Many thanks for the reviewer's valuable comment. We also agree that the expression of energy density is more accurate when calculated based on the entire mass of cell. Moreover, we believe that the aforementioned representation can give our result an edge over others. In addition, the nominal voltage throughout the cycling test was calculated, and the result showed that the nominal voltages of DPCE pouch cell with varying areal mass loadings were around 3.9 V as shown below. Therefore, this value was used for the calculation of energy densities.

Consequently, we recalculated the specific energy and volumetric energy density values based on the weight and volume of the accrual cell (including the pouch cell package) and provided the information in **Fig. 6e,f** and **Supplementary Table 1** of the revised manuscript as shown below. Note that additional pouch cell performance results with various other areal mass loading DPCEs have also been included in the revised manuscript (the calculation result in **Supplementary Table 1** represents the energy density of the Li-metal pouch cell assembled with the highest areal mass loading DPCE (100 mg cm⁻²)).

[Revised manuscript]

Fig. 6 | The ultrahigh mass loading DPCE (Pouch-cell). **a**, Charge/discharge voltage profile of the Li-metal DPCE pouch cell (31 mg cm^{-2}) at a voltage range of 2.9-4.3 V and charge/discharge current density of 0.1/0.1 C (dashed line represents the theoretical capacity of NCM712). **b**, Cycling performance of the Li-metal DPCE pouch cell (31 mg cm^{-2}) at a charge/discharge current

density of 0.5/0.5 C at a voltage range of 2.9-4.3 V. **c**, Charge/discharge voltage profiles of the Li-metal DPCE pouch cells in terms of areal mass loadings at a voltage range of 2.9-4.3 V and charge/discharge current density of 0.05/0.1 C. **d**, Cycling performance of the Li-metal DPCE pouch cells with various areal mass loadings. **e**, Areal capacities of cells as a function of areal mass loading (DPCEs versus previously reported solvent-free cathodes). **f**, Specific energies (calculated based on the entire cell weight) of cells as a function of areal capacity (DPCEs versus previously reported high mass loading cathodes).

“Given this ultrahigh loading design, a high energy density Li-metal pouch cell can be fabricated, and the calculated specific energy and volumetric energy density (based on the entire cell weight) were up to 360 Wh kg⁻¹ and 701 Wh L⁻¹, respectively at SOC 0% (Supplementary Table 1). It is noteworthy that this ultrahigh loading capability of the DPCE pouch cell far exceeded that of the previously reported solvent-free electrodes in terms of both areal capacity and mass loading (Fig. 6e, Supplementary Table 2), and even surpassing other high loading cathodes fabricated with different methods (Fig. 6f, Supplementary Table 3).”

Supplementary Table 1 | Calculation details for the specific energies and volumetric energy densities of the Li metal cells equipped with the DPCEs.

The specific energy of the Li metal cell was calculated based on the following equation:

$$\begin{aligned} \text{Specific energy (Wh kg}^{-1}\text{)} &= \frac{\text{Energy}}{\text{Mass of cell}} = \frac{\frac{\text{Energy}}{\text{Area}}}{\frac{\text{Mass of cell}}{\text{Area}}} \\ &= \frac{\text{Nominal voltage} \times C/A}{M_{\text{cathode}}/A + M_{\text{anode}}/A + M_{\text{separator}}/A + M_{\text{electrolyte}}/A + M_{\text{package}}/A} \end{aligned}$$

where M_{cathode} , M_{anode} , $M_{\text{separator}}$, $M_{\text{electrolyte}}$ and M_{package} denote the mass of cathode (including the etched Al current collector (20 μm)), anode (comprised of Li metal (200 μm) and Cu current collector (18 μm)), PE separator, injected electrolyte, and **pouch-cell package** respectively (C and A refer to the capacity and area, respectively).

C/A [mAh cm ⁻²]	Nominal voltage [V]	M_{cathode}/A [mg cm ⁻²]	M_{anode}/A [mg cm ⁻²]	$M_{\text{separator}}/A$ [mg cm ⁻²]	$M_{\text{electrolyte}}/A$ [mg cm ⁻²]	M_{package}/A [mg cm ⁻²]	M_{total}/A [mg cm ⁻²]	Specific energy [Wh kg ⁻¹]
17.6	3.9	105.4	26.8	0.8	36.0	21.3	190.3	360

The volumetric energy density of the Li metal cell was calculated based on the following equation:

$$\text{Volumetric energy density (Wh L}^{-1}\text{)} = \frac{\text{Energy}}{\text{Thickness of cell}} = \frac{\text{Nominal voltage} \times \text{C/A}}{T_{\text{cathode}} + T_{\text{anode}} + T_{\text{separator}} + T_{\text{package}}}$$

where T_{cathode} , T_{anode} , $T_{\text{separator}}$ and T_{package} are the thickness of cathode, anode, PE separator and **pouch-cell package**, respectively.

C/A [mAh cm ⁻²]	Nominal voltage [V]	T_{cathode} [μm]	T_{anode} [μm]	$T_{\text{separator}}$ [μm]	T_{package} [μm]	T_{total} [μm]	Volumetric energy density [Wh L ⁻¹]
17.6	3.9	593	218	18	150	979	701

REVIEWER COMMENTS

Reviewer #1 (Remarks to the Author):

The authors have made proper revision according to the suggestions. The previous concern on the control sample, and the mass basis of energy density is addressed. I think the manuscript is now suitable for publishing on Nat Common.

Reviewer #2 (Remarks to the Author):

The authors have addressed the reviewers' comments and the manuscript can be published after minor revision.

In the response, the authors used 3.7 g/cm³ for NCM density, which is too low. NCM density is usually around 4.7 g/cm³.

It would be good to discuss the limitation of this method/electrode.

Reviewer comments

Reviewer #1:

The authors have made proper revision according to the suggestions. The previous concern on the control sample, and the mass basis of energy density is addressed. I think the manuscript is now suitable for publishing on Nat Common.

Response: We greatly appreciate the reviewer's valuable and positive comments.

Reviewer #2:

The authors have addressed the reviewers' comments and the manuscript can be published after minor revision. In the response, the authors used 3.7 g/cm³ for NCM density, which is too low. NCM density is usually around 4.7 g/cm³. It would be good to discuss the limitation of this method/electrode.

Response: We greatly appreciate the reviewer's valuable suggestions. First of all, we give our sincere apologies for confusing the reviewer with a mistakenly written true density value of NCM, and we would also like to thank the reviewer for pointing this out and giving us the opportunity to correct our mistake. As the reviewer mentioned, the true density value of 4.7 g/cm³ should have been used for NCM active material instead of 3.7 g/cm³ in the previous response. We thus recalculated the porosity of the high loading electrodes based on the correct NCM true density value using the following equation^{1,2}:

$$\varepsilon = 1 - \frac{M}{T} \left[\left(\frac{\phi_{AM}}{D_{AM}} \right) + \left(\frac{\phi_{CA}}{D_{CA}} \right) + \left(\frac{\phi_B}{D_B} \right) \right]$$

where ε is the porosity of the electrodes, M is the areal mass loading of the electrodes, T is the thickness of the electrode, Φ_{AM} , Φ_{CA} , and Φ_B are the mass fractions of active material (NCM712), conductive agent (MWNT), and binder (PVDF) of the electrode, respectively, and D_{AM} , D_{CA} , and D_B are the true densities of active material (NCM712), conductive agent (MWNT), and binder (PVDF) of the electrode, respectively. The true densities of NCM712, MWNT, and PVDF are 4.7, 1.7, and 1.7 g cm⁻³, respectively. The calculated porosity values are shown in the table below:

Electrode composition	NCM712/MWNT/PVDF – 80/15/5			
Areal mass loading (mg cm ⁻²)	22	25	28	30
Thickness (μm)	98	123	176	221
Porosity (%)	35	41	54	60

1. Moroni, R., Börner, M., Zielke, L., Schroeder, M., Nowak, S., Winter, M., Manke, I., Zengerle, R., & Thiele, S. Multi-Scale correlative tomography of a li-ion battery composite cathode. *Sci. Rep.* **6**, 30109 (2016).
2. Ebner, M., Geldmacher, F., Marone, F., Stampanoni, M., & Wood, V. X-Ray tomography of porous, transition metal oxide based lithium ion battery electrodes. *Adv. Energy Mater.* **3**, 845–850 (2013).
3. Hamed, H., Yari, S., D’Haen, J., Renner, F. U., Reddy, N., Hardy, A., & Safari, M. Demystifying charge transport limitations in the porous electrodes of lithium-ion batteries. *Adv. Energy Mater.* **10**, 2002492 (2020).
4. Cunha, R. P., Lombardo, T., Primo, E. N., & Franco, A. A. Artificial intelligence investigation of NMC cathode manufacturing parameters interdependencies. *Batteries Supercaps* **3**, 60–67 (2019).
5. Nguyen, TT., Demortière, A., Fleutot, B. et al. The electrode tortuosity factor: why the conventional tortuosity factor is not well suited for quantifying transport in porous Li-ion battery electrodes and what to use instead. *npj Comput. Mater.* **6**, 123 (2020).
6. Lu, X., Bertei, A., Finegan, D.P. et al. 3D microstructure design of lithium-ion battery electrodes assisted by X-ray nano-computed tomography and modelling. *Nat. Commun.* **11**, 2079 (2020).

The calculation result shows that porosity of the high loading electrodes varies from 35 to 60%, which indicates that the porosity of electrode with areal mass loading of 30 mg cm⁻² marginally exceeds the average NCM cathode porosity values (30-50%)^{3,4}. It is known that the increase in electrode porosity facilitates the mass transport across a thick electrode by lowering tortuosity⁵, however, it is also accompanied by an increase of active particle/electrolyte interface, which can result in increased side reactions. Moreover, high porosity can aggravate the connectivity of electron-conducting pathways, thereby undermining the rate capability of the cell⁶.

Hence, future efforts should focus on lowering the electrode porosity of high loading electrode by improving the packing density and providing better particle-to-particle and particle-to-substrate contact. We believe that these aforementioned points will offer crucial guidelines for the future developments and therefore it was included in the revised manuscript as below:

[Revised manuscript]

“Moreover, Li-metal pouch cells with various areal mass loadings were fabricated (Fig. 6c) to test the ultrahigh loading capability of the DPCE. The cycling test showed stable cycling performance, emphasizing the applicability of the DPCE at extremely high loading conditions (Fig. 6d). Unsurprisingly for such ultrahigh loading electrodes, the discharge capacity at a higher C-rate (0.5 C corresponding to 6.76 mA cm^{-2}) was relatively low (Supplementary Fig. 19), presumably due to a combined effects of high porosity and high current density. Future studies will be dedicated to improving electron percolation pathways and material contact by enhancing the homogeneity of particle packing and micro-patterning of current collector.”